# RETHINK MINI-BATCH GRADIENT: CASCADE MOMENTUM

## ABSTRACT

During foundation model training, mini-batch stochastic gradient descent alleviates memory constraints; however, the resulting increase in gradient variance induces sharp oscillations in the loss curve, slowing convergence. Conventional momentum algorithms overlook the limitation introduced by mini-batch training; their ideal assumption is that momentum propagates smoothly over time. Yet, in practice, momentum is almost restricted to gradients within a single epoch, so cross-epoch information is severely diminished and cannot continuously suppress oscillations. For the first time, we theoretically analyze the momentum degradation problem under mini-batch gradients. To address this, we propose **Cascaded Momentum**, which splits momentum into an **Inner momentum** that rapidly smooths mini-batch gradients within each epoch and an **Outer momentum** that accumulates historical gradient trends across epochs to provide inertial guidance to subsequent epochs. This two-level mechanism simultaneously attenuates noise and accelerates convergence with virtually no additional cost.

## 1 INTRODUCTION

Deep neural networks (DNNs) have grown from millions to hundreds of billions of parameters, unlocking unparalleled performance in vision, language, and multi-modal reasoning Vaswani et al. (2023). Training these over-parameterised models, however, is increasingly gated by $GPU/TPU$ memory capacity: a full-batch update can easily exceed the on-chip memory. Consequently, **mini-batch stochastic gradient descent** (SGD), processing just a few dozen or hundreds of samples per step, has become the practical backbone of large-scale learning pipelines Bottou (2010). By design, mini-batching trades a modest increase in statistical noise for a dramatic reduction in memory footprint You et al. (2017), enabling data-parallel throughput that keeps pace with modern hardware. Yet this apparent win introduces a hidden cost. Reducing batch size inflates the variance of the gradient estimator Reddi et al. (2018): each update is now guided by a noisy snapshot of the loss surface rather than the population gradient. As batch sizes shrink, the optimization trajectory begins to zig-zag, producing steep oscillations in the loss curve and hindering stable convergence Haruki et al. (2019). For very deep or highly non-convex objectives, e.g. transformers with residual connections He et al. (2016), these oscillations can derail training altogether, causing long plateaus or catastrophic divergence unless the learning rate is aggressively curtailed Shallue et al. (2019). The **core dilemma** is clear: memory efficiency at the price of high-frequency noise Goyal et al. (2018); Kim et al. (2023).

Since heavy-ballPolyak (1964) popularization in the DNNs era, momentum has been the optimizer's shock absorber Sutskever et al. (2013). By averaging past gradients, it filters out high-frequency fluctuations and amplifies reliable, low-variance directions Goyal et al. (2018), thereby accelerating progress on smooth manifolds. The theoretical benefit presumes, however, that the momentum buffer persists as a continuous time series. In practice, mini-batch training, the momentum buffer assigns exponential weight to the most recent gradients, and because gradients from adjacent batches within the same epoch are highly correlated and updated at a rapid cadence Li et al. (2019), the buffer is quickly saturated by this **short-term, strongly correlated** signal. When a new epoch begins, the momentum vector is already dominated by the flurry of updates from the end of the previous epoch Jiang et al. (2019). Consequently, earlier cross-epoch gradient information is swiftly drowned out, preventing the momentum from conveying meaningful long-range cues into the next epoch. Despite rapid progress in optimizer design, e.g., Adam Kingma & Ba (2017), RAdam Liu et al. (2021), and

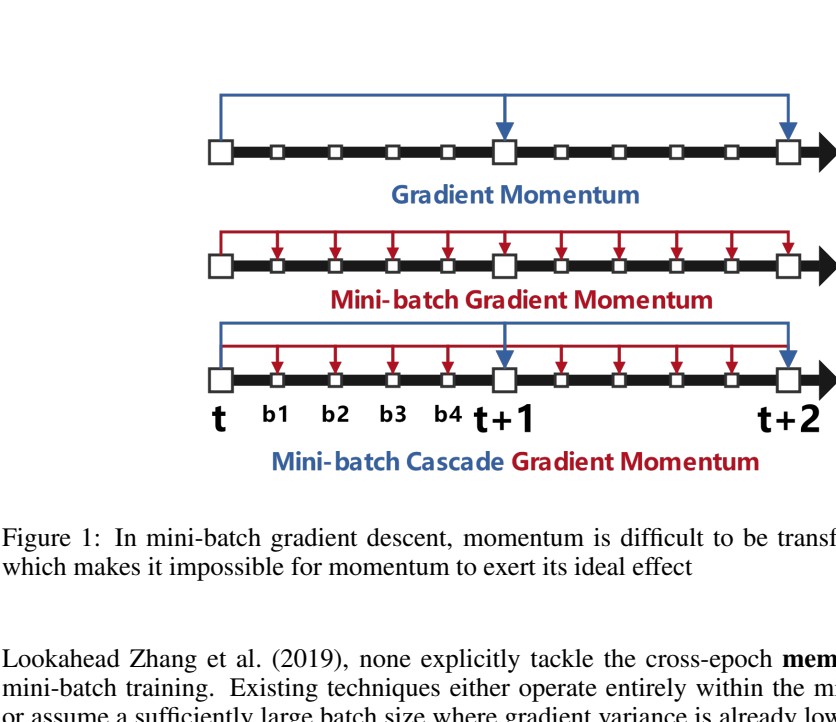

Figure 1: In mini-batch gradient descent, momentum is difficult to be transferred across epochs, which makes it impossible for momentum to exert its ideal effect

Lookahead Zhang et al. (2019), none explicitly tackle the cross-epoch **memory loss** induced by mini-batch training. Existing techniques either operate entirely within the micro-batch timescale, or assume a sufficiently large batch size where gradient variance is already low Goyal et al. (2018). Consequently, the community still lacks a lightweight, memory-neutral mechanism that preserves long-range momentum without sacrificing the benefits of mini-batches You et al. (2019).

In order to alleviate the above-mentioned challenges, we introduce **Cascaded Momentum (CM)**, a two-tier extension that restores long-term gradient memory without altering the simplicity or memory profile of standard SGD. "CM" decomposes the momentum accumulator into: **Inner momentum**: a fast, exponentially moving average that operates within an epoch, exactly mirroring classical momentum to suppress high-frequency noise among adjacent mini-batches. **Outer momentum**: a slow, persistent accumulator that spans epochs. After each epoch finishes, the final inner-momentum state is fed into the outer buffer, which then decays using a longer time constant. At the start of the next epoch, the outer buffer provides an initial velocity, effectively warm-starting the inner loop with historical context.

In summary, while mini-batch SGD is indispensable for memory-efficient training, its elevated gradient noise exposes a critical blind spot in classical momentum. Cascaded Momentum fills this gap with a theoretically grounded, practically effortless enhancement that marries local noise suppression to global trajectory guidance, bringing us one step closer to memory-efficient yet convergence-robust deep learning.

To evaluate the effectiveness of the "CM" method, we conduct both theoretical and experimental validations. Theoretically, we establish practical convergence guarantees for the traditional Stochastic Gradient Descent with Momentum (SGDM) algorithm and the proposed algorithm under mini-batch settings, followed by a comparative analysis. Experimentally, we train models using various SGDM variants under different batch size conditions. The results demonstrate that, consistent with the theoretical analysis, the SGDM enhanced by the "CM" mechanism consistently outperforms other SGDM variants across various batch size configurations after appropriate hyperparameter tuning. In summary, our main contributions are as follows:

- We propose SGD with Cascaded Momentum (SGD-CM), a novel algorithm designed to mitigate the degradation of cross-epoch momentum information caused by large-scale mini-batch training. SGD-CM significantly improves training stability across various batch-size configurations and accelerates the training process to some extent.

- Theoretically, we first refine the convergence analysis of the traditional SGD with momentum (often referred to as Normalized Stochastic Heavy Ball - NSHB) under the mini-batch setting. Our analysis aligns more closely with realistic scenarios compared to prior work, relies on milder assumptions, and yields tighter convergence rates. Furthermore, we establish the convergence guarantees for SGD-CM. Under certain conditions, we provide a comparative analysis of their convergence behaviors, demonstrating that the introduction of

Cascaded Momentum effectively alleviates the limitations inherent in traditional momentum methods.

- Through extensive experimentation, we empirically validate the efficacy of the proposed algorithm. The results demonstrate that SGD-CM attains nearly superior overall performance compared to other representative SGD with momentum variants., maintaining this competitive performance under diverse batch-size settings.

## 2 RELATED WORK

While momentum methods like Polyak's SHB Polyak (1964) and Nesterov's NAG Nesterov (1983) underlie accelerated optimization, theoretical understanding of SGDM in non-convex settings remains limited. Existing analyses Yan et al. (2018); Gitman et al. (2019) overlook how batch size affects momentum dynamics, a critical gap given that large-scale mini-batch processing submerges momentum information across epochs. Our **Cascade Momentum (CM)** addresses this by preserving long-term guidance while maintaining mini-batch efficiency. (Full discussion in Appendix A.)

## 3 METHOD

### 3.1 PROBLEM STATEMENT AND MOTIVATION

Considering the training problem of deep neural networks (DNNs), let $\theta \in \mathbb{R}^d$ represent the trainable parameters (weight matrices) of the DNN, where $\mathbb{R}^d$ is a $d$-dimensional Euclidean space endowed with inner product $\langle \cdot, \cdot \rangle$ and induced norm $\| \cdot \|$ Gao et al. (2021). For a given training set $\mathcal{D}_n = \{(x_1, y_1), \ldots, (x_n, y_n)\}$, the sample loss function is defined as $f_i(\theta) := l(\theta; (x_i, y_i)) : \mathbb{R}^d \to \mathbb{R}_+$. Then, the empirical risk minimization (ERM) problem can be expressed as:

$$\min_{\theta \in \mathbb{R}^d} \left[ f(\theta) := \frac{1}{n} \sum_{i=1}^{n} f_i(\theta) \right] \tag{1}$$

This finite sum optimization problem (FSOP) is a fundamental paradigm in machine learning and is widely found in logistic regression and DNN training Shalev-Shwartz & Ben-David (2014). When the number of parameters $d$ and the number of samples $n$ are large, traditional gradient descent (GD) methods face significant computational bottlenecks due to the need to calculate the full gradient Nesterov (2018). Stochastic gradient descent (SGD) addresses this issue by estimating the stochastic gradient of a batch of samples.

$$\theta_{k+1} = \theta_k - \eta_t \frac{1}{|\mathcal{b}_k|} \sum_{i \in \mathcal{b}_k} \nabla f_{\xi_i}(\cdot) \tag{2}$$

Here, $\eta_t$ is the step size. Although Stochastic Gradient Descent (SGD) is computationally efficient and can escape some local minima, its high variance characteristic leads to a slow convergence rate and significant oscillations during the training process Gao et al. (2021). To enhance performance, the Stochastic Gradient Descent with Momentum (SGDM) algorithm is widely adopted Sutskever et al. (2013), where $\beta$ is the momentum coefficient and $\tilde{g}_k$ is the stochastic gradient. This paper focuses on the advanced variant of SGDM, the Normalized Stochastic Heavy Ball (NSHB) method, whose update rule is as follows:

$$\begin{cases} v_k = \beta v_{k-1} + (1 - \beta) \tilde{g}_k, \\ \theta_{k+1} = \theta_k - \eta_t v_k, \end{cases} \tag{3}$$

When using minibatch gradients $\tilde{g}_k = \frac{1}{|\mathcal{b}_k|} \sum_{i \in \mathcal{b}_k} \nabla f_{\xi_i}(\cdot)$ with $\mathcal{b}_k \subset \{1, \ldots, n\}, |\mathcal{b}_k| = b$. Crucially, $b$ governs the gradient variance $\mathrm{Var}(\tilde{g}_k) \leq \sigma^2/b$, and we observe that in a small-batch training environment, the batch size not only directly affects the variance of the gradient estimation but also leads to the gradual drowning of the momentum-preserved gradient information from the previous round in the current round's iterative training. This implies that in a mini-batch training environment, relying solely on a single momentum term to naively accumulate historical gradients may fail to effectively preserve the desired information Johnson & Zhang (2013).

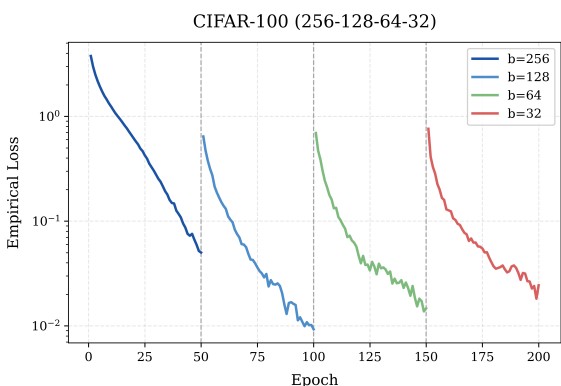

Figure 2: Empirical loss behavior of SGDM under decreasing multi-stage batch size training

To demonstrate this issue, we conduct a simple experiment using a ResNet-18 model on the CIFAR-100 dataset. The model is trained for 200 epochs with the stochastic gradient descent with momentum (SGDM) optimizer. We employ a decreasing batch size strategy ($256 \rightarrow 128 \rightarrow 64 \rightarrow 32$), dividing the training into four distinct phases. Figure 2 illustrates the empirical loss trajectory. A key observation reveals that the most pronounced loss increases coincide precisely with transitions between training phases, a phenomenon not observed during the stable periods within each phase. We attribute this behavior to mini-batch training dynamics. During stable phases, rapid iterative updates cause the gradient information preserved in the momentum buffer from previous stages to decay exponentially. As new gradients dominate the momentum term, the weight of historical gradients diminishes rapidly. Consequently, the momentum mechanism fails to provide effective long-term guidance. This ultimately accounts for both the decaying convergence rate and the loss oscillations observed in the later stages of conventional SGDM training.

To address the above problem, we have introduced the "CM" mechanism: (1) within a single round of iterative training, we maintain the original momentum form as the internal momentum,(2) we separately extract the internal momentum at the end of each round's iteration as the cumulative information of the external momentum, which is only passed between rounds; (3) in each round's iterative training, we use both the external and internal momenta to jointly guide the model's training process, thereby enhancing the model's training accuracy.

### 3.2 PROPOSED ALGORITHM

In this section, we first provide a brief introduction to the SGDM algorithm within the mini-Batch training setting, accompanied by a concise analysis of its operational procedure. Building upon this foundation, we introduce a novel algorithmic enhancement—SGDM augmented with the "CM" mechanism, offering a detailed exposition and analysis of its operational workflow.

---

**Algorithm 1:** SGD/ Momentum

---

1   **Input:** Dataset $\mathcal{B}$, momentum coefficient $\lambda$, momentum vector $v_i = 0$,learning rate $\eta_t$, Mini-batch size $b$.
2   **for** $t \in [T] = \{0, 1, 2...T-1\}$ **do**
3     $\theta_{t,0} \leftarrow \theta_t$
4     **for** $\mathcal{b}_k \in \mathcal{B}, k \in [K] = \{0, 1, 2...\lfloor \frac{\mathcal{B}}{b} \rfloor\}$ **do**
5       Compute gradient $\tilde{g}_{t,k} = \nabla f_{\mathcal{b}_k}(\theta_{t,k})$;
6       $m_1 = (1-\lambda)\tilde{g}_{t,k} + \lambda v_i$
7       $\theta_{t,k+1} = \theta_{t,k} - \eta_t m_1$
8       $v_i = \lambda v_i + (1-\lambda)\tilde{g}_{t,k}$;
9     **end**
10     $\theta_{t+1} \leftarrow \theta_{t,K}$;
11   **end**
12   **Return** $\theta_{T-1}$

---

In stochastic optimization, **Stochastic Gradient Descent with Momentum (SGDM)** accelerates convergence by accumulating velocity in directions of persistent gradient descent. The **Normalized Stochastic Heavy Ball (NSHB)** variant further enhances this by incorporating a lookahead gradient evaluation. For a mini-batch $\mathcal{b}_k$ of size $b$ sampled at iteration $t$, NSHB operates through four key steps: gradient computation $\tilde{g}_{t,k} = \nabla f_{\mathcal{b}_k}(\theta_{t,k})$, momentum integration $m_1 = (1-\lambda)\tilde{g}_{t,k} + \lambda v_i$, parameter update $\theta_{t,k+1} = \theta_{t,k} - \eta_t m_1$, and velocity propagation $v_i = \lambda v_i + (1-\lambda)\tilde{g}_{t,k}$, where $\lambda \in (0,1)$ is the momentum coefficient, $\eta_t > 0$ is the learning rate, and $v_i$ denotes the persistent velocity vector.

---

**Algorithm 2:** SGD/ Cascade / Momentum

1 **Input:** Dataset $\mathcal{B}$, Internal and external momentum coefficient $\lambda, \kappa$, Internal and external momentum vector $v_i, v_e = 0$, learning rate $\eta_t$.
2 **for** $t \in [T] = \{0, 1, 2...T-1\}$ **do**
3      $v_i^0 \leftarrow 0; \quad \theta_{t,0} \leftarrow \theta_t$
4      **for** $\mathcal{b}_k \in \mathcal{B}, k \in [K] = \{0, 1, 2... \lfloor \frac{\mathcal{B}}{b} \rfloor\}$ **do**
5          Compute gradient $\tilde{g}_{t,k} = \nabla f_{\mathcal{b}_k}(\theta_{t,k})$;
6          $m_1 = (1-\lambda)\tilde{g}_{t,k} + \lambda v_i^{k-1}$
7          $m_2 = (1-\kappa)m_1 + \kappa v_e^t$
8          $\theta_{t,k+1} = \theta_{t,k} - \eta_t m_2$
9          $v_i^k = \lambda v_i^{k-1} + (1-\lambda)\tilde{g}_{t,k}$;
10      **end**
11      $\theta_{t+1} \leftarrow \theta_{t,K}; \quad v_e^{t+1} = \kappa v_e^t + (1-\kappa)v_i^K$
12 **end**
13 **Return** $\theta_{T-1}$

---

A central innovation of our method lies in its **dual-momentum architecture**, designed to address limitations of single-buffer momentum in stochastic optimization. Classical momentum methods maintain a single momentum vector updated per mini-batch. However, this approach discards epoch-level gradient trends upon resetting buffers, potentially losing valuable inter-epoch directional information. To overcome this, we propose a **Cascaded Momentum framework** with two synergistic components: **Internal Momentum** ($v_i$): Updated per mini-batch to capture intra-epoch dynamics. **External Momentum** ($v_e$): Updated per epoch to preserve inter-epoch trajectory memory. The parameter update synthesizes both momentum components:

$$\theta_{t,k+1} = \theta_{t,k} - \eta_t \left[ (1-\kappa) \underbrace{\left((1-\lambda)\tilde{g}_{t,k} + \lambda v_i^{k-1}\right)}_{m_1} + \kappa v_e^t \right], \quad (4)$$

where $m_1$ represents the internal smoothed gradient. Crucially, $v_i$ is reset per epoch (Algorithm 2, line 3), while $v_e$ accumulates the final internal momentum of each epoch (Algorithm 2, line 11):

$$v_e^{t+1} \leftarrow \kappa v_e^t + (1-\kappa)v_i^K. \quad (5)$$

This design ensures $v_e$ retains a low-variance estimate of the epoch's directional consensus, acting as a **cross-epoch stabilizer**. When mini-batch gradients exhibit high volatility (e.g., in noisy or sparse landscapes), the external momentum provides persistent guidance, reducing zig-zagging and accelerating convergence. Conversely, the internal momentum adapts rapidly to local geometry within each epoch. This synergistic momentum cascade yields three significant algorithmic advantages:

- **Convergence Efficiency**: The external momentum's long-term memory mitigates epoch-boundary information loss, enabling steadier descent directions.

- **Implementation Simplicity**: Integration requires only two additional momentum buffers and one epoch-level update, a minimal change to existing optimizers.

- **Computational Economy**: Maintains $\mathcal{O}(d)$ memory and $\mathcal{O}(Kd)$ per-iteration cost (for $K$ mini-batches), matching standard momentum-SGD.

Consider a flat region where mini-batch gradients oscillate: $v_i$ smooths intra-epoch noise, while $v_e$ sustains progress direction across epochs. This cascade effect is analogous to a "momentum relay"; short-term adjustments refine long-term trajectory, and vice versa.

## 4 THEORETICAL ANALYSIS

In this section, we conduct an analysis of the convergence results of Algorithm 1 and 2 under non-convex conditions. To present the analysis, we begin by formulating the following assumption.

### 4.1 ASSUMPTIONS AND NOTATIONS

Throughout this convergence analysis, we employ the following mathematical notations: $f : \mathbb{R}^d \to \mathbb{R}$ denotes the objective function to minimize, with $f^*$ representing its global minimum value. Model parameters at iteration $t$ are denoted by $\theta_t \in \mathbb{R}^d$, where $\theta_0$ signifies the initial parameter vector. The full gradient is expressed as $\nabla f(\cdot)$, while $\nabla f(\cdot; \xi)$ indicates the stochastic gradient with respect to sample $\xi$. The expectation operator $\mathbb{E}[\cdot]$ applies to all random variables, and $\| \cdot \|$ denotes the standard Euclidean ($\ell_2$) norm. Key constants include $L$, the Lipschitz smoothness parameter from Assumption 1, and $\sigma^2$, the stochastic gradient variance bound defined in Assumption 2.

Optimization hyperparameters comprise the learning rate $\eta_t$ at iteration $t$, the internal momentum coefficient $\lambda$, and the external momentum coefficient $\kappa$ specific to SGD-CM. Data characteristics are captured by $b$ (mini-batch size), $B$ (total dataset size, serving as reference batch size), and $T$ (total number of training iterations). Asymptotic behavior is characterized by $\mathcal{O}(\cdot)$ for upper bounds and $\Theta(\cdot)$ for tight bounds.

**Assumption 1.** *$f : \mathbb{R}^d \to \mathbb{R}$ is L-smooth and differentiable, there exists $L > 0$ such that, for all $x, y \in \mathbb{R}^d$, satisfy that $\|\nabla f(x) - \nabla f(y)\| \leq L\|x - y\|$ .*

**Assumption 2.** *Let $\xi$ be an independent random variable, and let $\mathbb{E}_\xi$ denote expectation with respect to $\xi$. The stochastic gradient of $f$ at point $x$ is denoted by $\nabla f(x; \xi)$. The following assumptions hold: (i) **Unbiasedness**: $\mathbb{E}_\xi[\nabla f(x; \xi)] = \nabla f(x)$ for all $x \in \mathbb{R}^d$. (ii) **Mean Bounded Variance**: There exist constants $\Theta, \sigma \in [0, \infty)$ such that for all $x \in dom(f)$: $\frac{1}{B} \sum_{i=1}^{B} \|\nabla f(x; \xi_i) - \nabla f(x)\|^2 \leq \Theta\|\nabla f(x)\|^2 + \sigma^2$. The special case $\Theta = 0$ yields the conventional bounded variance assumption $\frac{1}{B} \sum_{i=1}^{B} \|\nabla f(x; \xi_i) - \nabla f(x)\|^2 \leq \sigma^2$, commonly adopted for non-convex objectives. Our formulation constitutes a strictly stronger condition that applies to a broader function class.*

**Assumption 3.** *Let $\mathcal{b}_k \in \mathcal{B}$ be a mini-batch consisting of $b$ independent and identically distributed (i.i.d.) random variables, denoted as $\mathcal{b}_k = (\xi_{k,1}, \xi_{k,2}, \cdots, \xi_{k,b})$. Let $g_{t,k}$ denote the full gradient of $f$ at $\theta_{t,k}$, and $\tilde{g}_{t,k}$ denote the stochastic gradient estimator. The stochastic gradient is computed as: $\tilde{g}_{t,k} = \nabla f(\theta_{t,k}; \mathcal{b}_k) = \frac{1}{b} \sum_{i=1}^{b} \nabla f(\theta_{t,k}; \xi_{k,i})$ and satisfies $\mathbb{E}_{\mathcal{b}_k}[\tilde{g}_{t,k}] = g_{t,k}$.*

### 4.2 CONVERGENCE ANALYSIS OF SGD/MOMENTUM WITH MINI-BATCH

First, we analyze the convergence of SGDM-Mini-batch in the non-convex case and present the results. See Appendix D for a detailed proof.

**Theorem 1.** *Suppose Assumption 1, 2 and 3 all hold. If the learning rate satisfies $\eta_t = \frac{\eta b}{B} \leq \frac{b}{\sqrt{6}BL}$ and $\eta^2 \leq \frac{1}{4L^2(\Theta + 2C)}$, where $C = \frac{4b^2\lambda^2}{B^2(1-\lambda)^2} + 1$, then the convergence result of SGDM-Mini-batch satisfies the following:*

$$\frac{1}{T} \sum_{t=0}^{T-1} \mathbb{E}\left[\|g_t\|^2\right] \leq \frac{4}{\eta T}\left[f(\theta_0) - f^*\right] + 4\eta^2 L^2 (4\lambda^2 + 3)\frac{\sigma^2}{b}. \tag{6}$$

The optimization term $\frac{4}{\eta T}[f(\theta_0) - f^*])$ constitutes an intrinsic component of standard (stochastic) gradient descent with momentum (SGD/M) optimizers. It fundamentally represents the primary driving force of the optimization process, quantifying the algorithm's capacity and efficiency in progressing from the initial point $\theta$ towards the global minimum $f^*$ of the objective function. The presence and form of this term are universally observed in deterministic gradient descent (GD) or idealized full-batch SGD. Its decaying property (asymptotically approaching zero with increasing iteration count $T$) provides the foundational guarantee for optimizer convergence.

In contrast, the noise term $4\eta^2 L^2 (4\lambda^2 + 3)\frac{\sigma^2}{b}$ emanates specifically from the mini-batch training paradigm. It arises directly from the practice of computing the stochastic gradient estimate $\tilde{g}_{t,k}$

using a mini-batch $b$, rather than the true gradient $g_{t,k}$. Since each mini-batch represents only a random subset of the underlying data distribution, the computed gradient inherently incorporates stochastic perturbations, or "noise". The variance $\sigma^2$ of this noise is the core source of this term. It induces oscillations in the optimization trajectory and ultimately manifests as a steady-state error, fundamentally limiting the algorithm's ability to reach the precise optimal solution. The magnitude of the noise term is inversely proportional to the mini-batch size $b$ (larger $b$ yields more accurate estimates and lower noise). Critically, however, for any finite $b$ (i.e., whenever stochastic gradients are employed), this term persists. Thus we can know the convergence rate of SGDM in the case of mini-batch as $\mathcal{O}(\sqrt{\frac{f(\theta_0)-f^*}{\eta T} + \frac{(4\lambda^2+3)\eta^2 L^2 \sigma^2}{b}})$.

In order for the upper bound in Theorem 1 to approach zero infinitely as $T$ increases, we need to additionally make some requirements on the learning rate $\eta$ that guarantee stable convergence.

**Corollary 1.** *Suppose the conditions in Theorem 1 are satisfied. If choosing the learning rates $\eta = \Theta(T^{-\frac{1}{2}})$, then the convergence result of SGDM-Mini-batch satisfies:*

$$\frac{1}{T} \sum_{t=0}^{T-1} \mathbb{E}\left[\|g_t\|\right] = \mathcal{O}\left(\sqrt{\frac{1}{T^{1/2}} + \frac{(4\lambda^2+3)\sigma^2}{Tb}}\right). \tag{7}$$

*From Corollary 1, we can know that when the number of training rounds $T$ is large enough, the SGDM algorithm can still achieve the convergence rate of $\mathcal{O}(\frac{1}{T^{1/4}})$ in the case of mini-batch. And when we increase the batch size $b$, the noise generated by the stochastic gradient will gradually decay, so that the convergence rate will be significantly improved, which is consistent with the situation we described before. Moreover, it is evident from the result that after setting the learning rate to $\Theta(T^{-\frac{1}{2}})$, the convergence rate of the overall expression is dominated by the optimization term $\frac{f(\theta_0)-f^*}{\eta T}$. However, the momentum coefficient $\lambda$ primarily influences the noise term, which implies that adjusting $\lambda$ cannot consistently achieve stable convergence rates across different batch sizes.*

Compared with prior work, we contrast the convergence results of our SGDM method under non-convex objectives and mini-batch settings, primarily against those of Liu et al. (2020) and Liang et al. (2023). However, Liang et al. (2023) primarily targets a specific class of non-convex objective functions satisfying the Polyak–Łojasiewicz (PL) condition, and thus their analysis remains confined to this scenario, presenting certain limitations. Regarding the underlying assumptions, neither our work nor the compared studies rely on overly strong assumptions, such as the Bounded Gradient hypothesis. For the dominant optimization term $\mathcal{O}\left(\frac{f(\theta_0)-f^*}{\eta T}\right)$, we achieve results matching Liu et al. (2020), while we achieve a tighter bound on the noise term: our $\frac{(4\lambda^2+3)\eta^2 L^2 \sigma^2}{b}$ improves upon $\mathcal{O}\left(L\eta\sigma^2\right)$ in Liu et al. (2020). Finally, we further analyze the influence of the momentum coefficient $\lambda$ on the overall convergence of the algorithm under a gradually decaying learning rate schedule.

### 4.3 Convergence Analysis of SGD/Cascade Momentum with Mini-batch

Next we analyze the convergence results of the SGDM-Mini-batch algorithm with the "CM" mechanism in the non-convex setting, and see the Appendix C for the detailed proof.

**Theorem 2.** *Suppose Assumption 1, 2 and 3 all hold. If the learning rate satisfies: $\eta_t \leq \min\left\{\frac{1-\lambda}{4L[(4+\lambda^2)+2(1-\lambda)(1-\kappa)]}, \frac{1-\lambda}{2\sqrt{2}L(1-\kappa)\sqrt{\lambda+\lambda^2}}\right\}$, then the convergence result of SGDM-Mini-batch with the "CM" mechanism satisfies the following:*

$$\frac{1}{T} \sum_{t=0}^{T-1} \mathbb{E}\left[\|g_t\|^2\right] \leq \frac{8L\eta_t\sigma^2}{b} + \frac{1-\lambda}{1-8\kappa^2\frac{B}{b}(1-\lambda)} \tag{8}$$
$$\cdot \left(\frac{2(f(\theta_0)-f^*)}{\eta_t} + \frac{B}{b}\frac{8L\eta_t\sigma^2}{b} + \frac{1-\kappa}{(1+\lambda)(1+\kappa)}\frac{\sigma^2}{b}\right).$$

For this upper bound, we should first ensure that the denominator of the coefficient of the second term on the right is always positive, that is $\kappa^2(1-\lambda) \leq \frac{1}{8\frac{B}{b}}$ is satisfied. Similarly, in order to ensure that the upper bound can approach zero when $T$ is large enough, we split the

right-hand side of the above equation into four terms $A\eta_t + \frac{1-\lambda}{1-8\kappa^2 \frac{B}{b}(1-\lambda)} \left[ B\frac{1}{\eta_t} + C\eta_t + D \right]$, where $A = \frac{8L\sigma^2}{b}$, $B = 2[f(\theta_0) - f^*]$, $C = \frac{8L\sigma^2}{b}\frac{B}{b}$, $D = \frac{(1-\kappa)\sigma^2}{(1+\lambda)(1+\kappa)b}$. Thus we can know the convergence rate of SGD-CM in the case of mini-batch as $\mathcal{O}(\frac{1-\lambda}{1-8\kappa^2 \frac{B}{b}(1-\lambda)} \left[ \frac{f(\theta_0)-f^*}{\eta_t} \right] +$ $\left[ L\eta_t \left(1 + \frac{1-\lambda}{1-8\kappa^2 \frac{B}{b}(1-\lambda)}\right) + \frac{1-\lambda}{1-8\kappa^2 \frac{B}{b}(1-\lambda)} \frac{1-\kappa}{(1+\lambda)(1+\kappa)} \right] \frac{\sigma^2}{b})$.

**Corollary 2.** *Suppose the conditions in Theorem 2 are satisfied. If choosing the learning $\eta_t = \Theta(T^{-\frac{1}{2}})$ and $1 - \lambda = \Theta(T^{-1})$, we can obtain $\frac{1-\lambda}{\eta_t} = \Theta(T^{-\frac{1}{2}})$ and $\kappa^2 \leq \frac{b}{8(1-\lambda)B}$, then the convergence result of SGD-CM-Mini-batch satisfies:*

$$\frac{1}{T}\sum_{t=0}^{T-1} \mathbb{E}\left[\|g_t\|\right] = \mathcal{O}\left( \sqrt{\frac{1}{1-8\kappa^2\frac{B}{b}(1-\lambda)} T^{-\frac{1}{2}} + \frac{\sigma^2}{b} T^{-\frac{1}{2}}} \right. \tag{9}$$
$$\left. + \sqrt{\frac{1}{1-8\kappa^2\frac{B}{b}(1-\lambda)} \left( T^{-\frac{3}{2}} + T^{-1}\frac{1-\kappa}{(1+\lambda)(1+\kappa)} \right) \frac{\sigma^2}{b}} \right).$$

*Corollary 2 establishes that SGD-CM maintains an asymptotic convergence rate of $\mathcal{O}(\frac{1}{T^{1/4}})$ in mini-batch settings—a property consistent with classical SGDM (Corollary 1). However, the fundamental distinction lies in its convergence robustness to batch size variations: although both methods can accelerate convergence to some extent by increasing the batch size $b$, SGD-CM uniquely preserves this advantage through its batch-dependent momentum coordination mechanism governed by $\frac{1-\lambda}{1-8\kappa^2\frac{B}{b}(1-\lambda)}$. Specifically, the dominant term $\frac{f(\theta_0)-f^*}{\eta T}$ in Corollary 1 corresponds directly to the $\frac{1-\lambda}{1-8\kappa^2\frac{B}{b}(1-\lambda)} \left[ \frac{f(\theta_0)-f^*}{\eta_t} \right]$ component within the expression $\frac{1-\lambda}{1-8\kappa^2\frac{B}{b}(1-\lambda)} \left[ \frac{f(\theta_0)-f^*}{\eta_t} \right] + \left[ L\eta_t \left(1 + \frac{1-\lambda}{1-8\kappa^2\frac{B}{b}(1-\lambda)}\right) + \frac{1-\lambda}{1-8\kappa^2\frac{B}{b}(1-\lambda)} \frac{1-\kappa}{(1+\lambda)(1+\kappa)} \right] \frac{\sigma^2}{b}$, whose coefficient structure remains stable under the constraint $\kappa^2(1-\lambda) < \frac{1}{8}\frac{b}{B}$. This structural consistency explicitly achieves the core objective of introducing external momentum.*

The convergence behavior induced by the CM mechanism exhibits fundamental distinctions compared to conventional SGDM algorithms. The most significant divergence manifests as a structural transformation in the dominant terms of the convergence rate. Specifically, Theorem 2 reveals that the critical convergence component $\frac{2(f(\theta_0)-f^*)}{\eta_t} + \frac{B}{b}\frac{8L\eta_t\sigma^2}{b} + \frac{1-\kappa}{(1+\lambda)(1+\kappa)}\frac{\sigma^2}{b}$ no longer depends exclusively on the learning rate $\eta_t$. Instead, it demonstrates strong dual dependence on both momentum coefficients $\lambda$ and $\kappa$. This interdependence is primarily governed by the scaling factor $\frac{1-\lambda}{1-8\kappa^2\frac{B}{b}(1-\lambda)}$. Consequently, coordinated adjustment of both momentum hyperparameters becomes essential for controlling the convergence behavior. Notably, the tuning of the external momentum coefficient $\kappa$ exhibits batch-size dependency, as quantified by the constraint: $\kappa^2(1-\lambda) < \frac{1}{8}\frac{b}{B}$. This intrinsic property enables SGD-CM to maintain convergence robustness across varying batch sizes, which aligns precisely with the original design intention behind introducing external momentum.

## 5 EXPERIMENTS

For the experimental evaluation, we compared not only common variants of the SGDM algorithm, namely Stochastic Heavy Ball (SHB) and Nesterov Stochastic Heavy Ball (NSHB), but also the Adam, AdamW, and the base SGD optimizer. Experiments were conducted using a ResNet-18 network on both the CIFAR-100 and Tiny ImageNet datasets across various batch-size configurations. Hardware Environment: All experiments were performed on a system equipped with an NVIDIA A40 GPU and an Intel Xeon Gold 5318Y CPU. Software Environment: We utilized Python 3.12.2, PyTorch 2.3.1, and CUDA 12.2.

Training Configuration: Each optimizer was trained for 200 epochs under batch sizes of 32, 64, 128, and 256. Hyperparameters for all optimizers followed recommended default settings, and for our proposed SGD-CM algorithm, parameters were configured according to the theoretically derived optimal settings. SGD-CM:$lr = \frac{0.1}{\sqrt{T}}, \lambda = 1 - \frac{1}{T}, \kappa = \frac{1}{\sqrt{8B/b}}$; Adam/AdamW: $lr = 0.001, betas = (0.9, 0.999)$; SGD/SHB/NSHB: $lr = 0.1, beta = 0.9$.

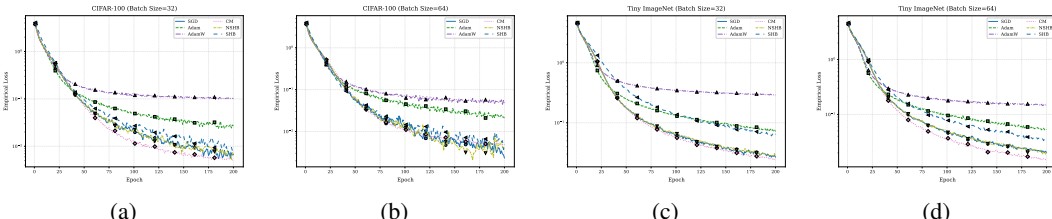

(a)           (b)           (c)           (d)

Figure 3: Comparison of empirical loss across optimizers: (a-b) CIFAR-100 with ResNet-18, (c-d) Tiny ImageNet with ResNet-18. Batch sizes (BS) are 32 and 64 as indicated.

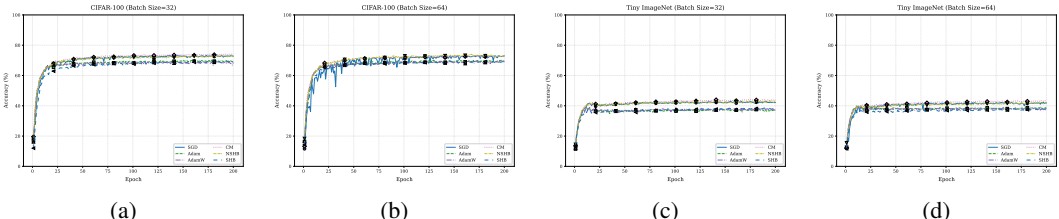

(a)           (b)           (c)           (d)

Figure 4: Comparison of test accuracy across optimizers and datasets: (a) CIFAR-100 with batch size 32, (b) CIFAR-100 with batch size 64, (c) Tiny ImageNet with batch size 32, (d) Tiny ImageNet with batch size 64. All experiments use ResNet-18 architecture.

Figures 3 and 4 present comparisons of empirical loss curves and test accuracy curves, respectively, for various optimizer algorithms trained on the CIFAR-100 and Tiny ImageNet dataset using ResNet-18 under different batch-size settings (32 or 64). Correspondingly. Regarding loss curves, the SGD-CM algorithm demonstrates superior stability compared to all other algorithms on the CIFAR-100 and Tiny ImageNet datasets.

Furthermore, it achieves the best performance under smaller batch-size settings (32 or 64) on both datasets. In terms of accuracy, our algorithm also outperforms others under smaller batch sizes. However, it is noteworthy that its accuracy performance is less competitive under larger batch sizes (128 or 256). We hypothesize that this limitation may stem from the algorithm's lack of a smooth transition between SGD-like and GD-like behaviors. Theoretically, increasing batch size should enhance convergence speed, but achieving this with SGD-CM likely requires additional considerations, such as jointly adjusting the external and internal momentum coefficients. However, an intriguing experimental finding is that, through appropriate adjustment of the momentum coefficients $\lambda$ and $\kappa$, we can also make SGD-CM achieve its best performance under larger batch-size settings (128 or 256). These aspects represent potential directions for future work.

## 6 CONCLUSION

In this paper, we propose an improved SGDM algorithm incorporating a Cascade Momentum mechanism. By decoupling traditional momentum into external momentum and internal momentum, our method achieves long-term memory of momentum information across epochs within the mini-batch training paradigm. For non-convex objectives, we mathematically enhance the convergence analysis of the conventional SGDM algorithm and provide a rigorous convergence guarantee for SGD-CM. We further prove that by jointly adjusting the internal and external momentum coefficients, our algorithm ensures stable convergence across varying batch-size settings. Experimental results demonstrate that SGD-CM delivers superior performance under smaller batch sizes. Our approach provides a practical guideline for enhancing SGDM in mini-batch training environments, namely, exploring alternative forms of momentum delivery during practical optimization.

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

## A    RELATED WORK

The theoretical foundations of accelerated gradient methods trace back to classical momentum: Polyak's Stochastic Heavy-Ball (SHB) Polyak (1964) and Nesterov's Accelerated Gradient (NAG) Nesterov (1983). NAG employs a look-ahead update strategy, achieving the theoretically optimal convergence rate for convex optimization. Building on this, Sutskever et al. (2013) integrated Polyak momentum into Stochastic Gradient Descent (SGD) for neural network training and proposed a practical heuristic implementation of Nesterov momentum. Owing to its empirical advantages in convergence speed and generalization, SHB has become the de facto standard in practice.

Despite empirical success, theoretical convergence guarantees for SGD with Momentum (SGDM) in non-convex settings remain incomplete. Early analyses focused primarily on the Normalized SHB (NSHB) variant Gupal & Bazhenov (1972). Yan et al. (2018) addressed this gap by establishing a unified framework for analyzing SHB and SGDM convergence under non-convexity. They proved an $O(1/\sqrt{T})$ convergence rate and quantified the relationship between the momentum coefficient $\beta$ and convergence speed. Gitman et al. (2019) further demonstrated that Polyak momentum reduces gradient noise via exponential smoothing, explaining its generalization benefits.

However, these works overlooked two critical aspects: (1) the effect of batch size on momentum dynamics, and (2) curvature-driven oscillations in complex loss landscapes. Recent studies confirm batch size fundamentally alters optimization dynamics: large batches converge to sharp minima with poor generalization Keskar et al. (2017), while small batches benefit from gradient noise acting as implicit regularization Jastrzębski et al. (2018).

Furthermore, the optimal learning rate for adaptive optimizers exhibits complex relationships with batch size due to noise-curvature transitions Thomas et al. (2020). Convergence analysis has also advanced for more complex scenarios. Mai & Johansson (2021) provided convergence proofs for SGDM applied to non-smooth, non-convex problems (e.g., networks with ReLU activations), introducing a generalized smoothness assumption to replace traditional Lipschitz continuity Zhang et al. (2020).

Yu et al. (2019) proved that distributed SGDM achieves linear speedup in non-convex optimization—where convergence speed increases with the number of workers—and provided convergence guarantees under communication compression Karimireddy et al. (2019). Liu et al. (2020) improved existing convergence bounds, showing SGDM achieves an $O(1/\sqrt{T})$ rate under a constant learning rate when the Polyak-Łojasiewicz (PL) condition holds, reducing sensitivity to learning rate tuning Khaled & Richtárik (2020).

However, these theoretical studies are not based on mini-batch training conditions and fail to account for the practical impact of batch size on noise variance, creating a theory-practice mismatch. This issue is particularly pronounced in SGDM-related algorithms, where empirical studies demonstrate that momentum coefficients must increase when batch sizes approach critical thresholds Shallue et al. (2019). In practice, each training round (epoch) involves iterative updates over numerous mini-batches. Consequently, the momentum information from the previous round becomes gradually submerged during massive mini-batch processing, hindering its intended guiding role. Therefore, we explore different methods for transferring momentum between round/epoch levels and propose a **"CM"** method. "CM" enables momentum to guide training within local epochs while preserving long-term guiding information across rounds, maintaining the advantage of mini-batch training.

## B    PROOF OF PRELIMINARY LEMMAS

**Lemma B.1.** *Under Assumption 2 and Assumption 3, the following holds:*

$$\mathbb{V}_{\mathcal{b}_k}\left(\tilde{g}_{t,k}\right) = \mathbb{E}_{\mathcal{b}_k}\left[\|\tilde{g}_{t,k} - g_{t,k}\|^2\right] \leq \frac{\sigma^2}{b}. \tag{10}$$

*Proof of lemma B.1.* From Assumption 2 (i) we can know that:

$$\mathbb{E}_{\mathcal{b}_k}\left[\tilde{g}_{t,k}\right] = \mathbb{E}_{\mathcal{b}_k}\left[\nabla f(\theta_{t,k}; \mathcal{b}_k)\right] = \frac{1}{b}\sum_{i=1}^{b}\mathbb{E}_{\xi_{k,i}}\left[\nabla f\left(\theta_{t,k}; \xi_{k,i}\right)\right] = \frac{1}{b}\sum_{i=1}^{b}g_{t,k} = g_{t,k},$$

this means that $\tilde{g}_{t,k}$ is an unbiased estimator of $g_{t,k}$, i.e:

$$\mathbb{E}_{\mathcal{b}_k}\left[\tilde{g}_{t,k} - g_{t,k}\right] = 0,$$

because $\mathcal{b}_k = (\xi_{k,1}, \xi_{k,2}, \cdots, \xi_{k,b})$ and $\xi_{k,1}, \xi_{k,2}, \cdots, \xi_{k,b}$ is independent and identically distributed,

$$\mathbb{V}_{\mathcal{b}_k}\left(\tilde{g}_{t,k}\right) = \mathbb{E}_{\mathcal{b}_k}\left[\|\frac{1}{b}\sum_{i=1}^{b}\nabla f\left(\theta_{t,k};\xi_{k,i}\right) - g_{t,k}\|^2\right]$$

$$= \frac{1}{b^2}\mathbb{E}_{\mathcal{b}_k}\left[\langle\sum_{i=1}^{b}\left[\nabla f\left(\theta_{t,k};\xi_{k,i}\right) - g_{t,k}\right], \sum_{j=1}^{b}\left[\nabla f\left(\theta_{t,k};\xi_{k,j}\right) - g_{t,k}\right]\rangle\right]$$

$$= \frac{1}{b^2}\left[\sum_{i=1}^{b}\mathbb{E}_{\mathcal{b}_k}\left[\|\nabla f\left(\theta_{t,k};\xi_{k,i}\right) - g_{t,k}\|^2\right] + \sum_{i\neq j}\mathbb{E}_{\mathcal{b}_k}\left[\langle\left[\nabla f\left(\theta_{t,k};\xi_{k,i}\right) - g_{t,k}\right], \left[\nabla f\left(\theta_{t,k};\xi_{k,j}\right) - g_{t,k}\right]\rangle\right]\right]$$

$$\overset{(a)}{=} \frac{1}{b^2}\sum_{i=1}^{b}\mathbb{E}_{\mathcal{b}_k}\left[\|\nabla f\left(\theta_{t,k};\xi_{k,i}\right) - g_{t,k}\|^2\right] \leq \frac{\sigma^2}{b}$$

Equation (a) holds because for any mutually independent random variables $\xi_{k,i}$ and $\xi_{k,j}$ $(i \neq j)$, and by Assumption 2(i), we have

$$\mathbb{E}_{\mathcal{b}_k}\left[\langle\left[\nabla f\left(\theta_{t,k};\xi_{k,i}\right) - g_{t,k}\right], \left[\nabla f\left(\theta_{t,k};\xi_{k,j}\right) - g_{t,k}\right]\rangle\right] = 0.$$

Furthermore, the last equation holds due to Assumption 2(ii).This completes the proof. $\square$

**Lemma B.2.** *Under the premises of Lemma B.1 and Algorithm 2, the following inequality holds:*

$$\mathbb{E}\left[\left\|v_{t,k} - (1-\lambda)\sum_{i=1}^{k}\lambda^{k-i}g_{t,i}\right\|^2\right] \leq \frac{1-\lambda}{1+\lambda}(1-\lambda^{2k})\frac{\sigma^2}{b}. \tag{11}$$

*Proof of Lemma B.2.*

$$\mathbb{E}\left[\left\|v_{t,k} - (1-\lambda)\sum_{i=1}^{k}\lambda^{k-i}g_{t,i}\right\|^2\right] \overset{(a)}{=} (1-\lambda)^2\mathbb{E}\left[\left\|\sum_{i=1}^{k}\lambda^{k-i}\left[\tilde{g}_{t,i} - g_{t,i}\right]\right\|^2\right]$$

$$= (1-\lambda)^2\mathbb{E}\left[\left\|\sum_{i=1}^{k}\lambda^{k-i}\left[\tilde{g}_{t,i} - g_{t,i}\right]\right\|^2\right]$$

$$= (1-\lambda)^2\mathbb{E}\left[\sum_{i=1}^{k}\sum_{j=1}^{k}\left\langle\lambda^{k-i}(\tilde{g}_{t,i} - g_{t,i}), \lambda^{k-j}(\tilde{g}_{t,j} - g_{t,j})\right\rangle\right]$$

$$= (1-\lambda)^2\sum_{i=1}^{k}\lambda^{2(k-i)}\mathbb{E}\left[\|\tilde{g}_{t,i} - g_{t,i}\|^2\right]$$

$$\overset{(b)}{\leq} (1-\lambda)^2\frac{\sigma^2}{b}\sum_{i=1}^{k}\lambda^{2(k-i)}$$

$$= \frac{1-\lambda}{1+\lambda}(1-\lambda^{2k})\frac{\sigma^2}{b},$$

where equation (a) follows from the definition of $v_{t,k}$ in Assumption 2:

$$v_{t,k} = \lambda v_{t,k-1} + (1-\lambda)\tilde{g}_{t,k}$$
$$= \lambda\left[\lambda v_{t,k-2} + (1-\lambda)\tilde{g}_{t,k-1}\right] + (1-\lambda)\tilde{g}_{t,k}$$
$$= \lambda\left[\lambda\left[\lambda v_{t,k-3} + (1-\lambda)\tilde{g}_{t,k-2}\right] + (1-\lambda)\tilde{g}_{t,k-1}\right] + (1-\lambda)\tilde{g}_{t,k}$$
$$= \cdots$$
$$= (1-\lambda)\sum_{i=1}^{k}\lambda^{k-i}\tilde{g}_{t,i}.$$

And equation (b) holds due to lemma B.1, This completes the proof. □

**Lemma B.3.** *We quote the following lemma from prior work:*

$$\mathbb{E}\left[\left\|\frac{1-\lambda}{1-\lambda^k}\sum_{i=1}^k \lambda^{k-i}g_{t,i} - g_{t,k}\right\|^2\right] \leq \sum_{i=1}^{k-1} a_{k,i}\mathbb{E}\left[\|\theta_{t,i+1}-\theta_{t,i}\|^2\right], \tag{12}$$

*where $a_{k,i} = \frac{L^2\lambda^{k-i}}{1-\lambda^k}\left(k-i+\frac{\lambda}{1-\lambda}\right)$.*

**Lemma B.4.** *Following the convergence analysis of SGDM in prior work, we define $z_{t,k}$ as:*

$$z_{t,k} = \begin{cases} \theta_{t,k} & k=1, \\ \dfrac{1}{1-\lambda}\theta_{t,k} - \dfrac{\lambda}{1-\lambda}\theta_{t,k-1} & k\geq 2. \end{cases} \tag{13}$$

*Furthermore, based on Algorithm 2, $z_{t,k+1} - z_{t,k}$ satisfies:*

$$z_{t,k+1} - z_{t,k} = -\eta_t\left[\kappa v_e^{t-1} + (1-\kappa)\tilde{g}_{t,k}\right].$$

*Proof of Lemma B.4.*

$$\begin{aligned}
z_{t,k+1} - z_{t,k} &= \frac{1}{1-\lambda}\theta_{t,k+1} - \frac{\lambda}{1-\lambda}\theta_{t,k} - \left(\frac{1}{1-\lambda}\theta_{t,k} - \frac{\lambda}{1-\lambda}\theta_{t,k-1}\right) \\
&= \frac{1}{1-\lambda}(\theta_{t,k+1} - \theta_{t,k}) - \frac{\lambda}{1-\lambda}(\theta_{t,k} - \theta_{t,k-1}) \\
&= \frac{1}{1-\lambda}\left[-\eta_t\left[\kappa v_e^{t-1} + (1-\kappa)v_{t,k}\right]\right] \\
&\quad - \frac{\lambda}{1-\lambda}\left[-\eta_t\left[\kappa v_e^{t-1} + (1-\kappa)v_{t,k-1}\right]\right] \\
&= -\eta_t\left[\kappa v_e^{t-1} + (1-\kappa)\left[\frac{1}{1-\lambda}(v_{t,k} - \lambda v_{t,k-1})\right]\right] \\
&= -\eta_t\left[\kappa v_e^{t-1} + (1-\kappa)\tilde{g}_{t,k}\right],
\end{aligned}$$

which completes the proof. □

## C PROOFS FOR THE CONVERGENCE OF SGD/CASCADE MOMENTUM WITH MINI-BATCH

*Proof of Theorem 1.* First, by the smoothness assumption of function $f$ and the definition of $z_{t,k}$, we have:

$$\mathbb{E}\left[f\left(z_{t,k+1}\right)\right] \leq \mathbb{E}\left[f\left(z_{t,k}\right)\right] + \mathbb{E}\left[\langle\nabla f\left(z_{t,k}\right), z_{t,k+1} - z_{t,k}\rangle\right] + \frac{L}{2}\mathbb{E}\left[\|z_{t,k+1} - z_{t,k}\|^2\right]$$

$$= \mathbb{E}\left[f\left(z_{t,k}\right)\right] + \underbrace{\mathbb{E}\left[\langle\nabla f\left(z_{t,k}\right), -\eta_t\left[(1-\kappa)\tilde{g}_{t,k} + \kappa v_e^{t-1}\right]\rangle\right]}_{S_0} + \underbrace{\frac{L\eta_t^2}{2}\mathbb{E}\left[\|(1-\kappa)\tilde{g}_{t,k} + \kappa v_e^{t-1}\|^2\right]}_{S_1}.$$

Next, we bound $S_0$ and $S_1$ separately. Starting with $S_0$:

$$S_0 = \mathbb{E}\left[\left\langle \nabla f\left(z_{t,k}\right), -\eta_t\left[(1-\kappa)\tilde{g}_{t,k} + \kappa v_e^{t-1}\right]\right\rangle\right]$$

$$= \mathbb{E}\left[\left\langle \nabla f\left(z_{t,k}\right) - g_{t,k}, -\eta_t\left[(1-\kappa)g_{t,k} + \kappa v_e^{t-1}\right]\right\rangle\right] - \eta_t(1-\kappa)\mathbb{E}\left[\|g_{t,k}\|^2\right] + \mathbb{E}\left[\left\langle g_{t,k}, -\eta_t\kappa v_e^{t-1}\right\rangle\right]$$

$$\overset{(a)}{\leq} \eta_t\frac{\rho_0}{2}L^2\mathbb{E}\left[\|z_{t,k} - \theta_{t,k}\|^2\right] + \eta_t\frac{1}{2\rho_0}\mathbb{E}\left[\|(1-\kappa)g_{t,k} + \kappa v_e^{t-1}\|^2\right] - \eta_t(1-\kappa)\mathbb{E}\left[\|g_{t,k}\|^2\right]$$

$$+ \eta_t\kappa\frac{\rho_1}{2}\mathbb{E}\left[\|g_{t,k}\|^2\right] + \eta_t\kappa\frac{1}{2\rho_1}\mathbb{E}\left[\|v_e^{t-1}\|^2\right]$$

$$= \eta_t\frac{\rho_0}{2}L^2\mathbb{E}\left[\|z_{t,k} - \theta_{t,k}\|^2\right] + \eta_t\frac{1}{\rho_0}(1-\kappa)^2\mathbb{E}\left[\|g_{t,k}\|^2\right] + \eta_t\frac{1}{\rho_0}\kappa^2\mathbb{E}\left[\|v_e^{t-1}\|^2\right] - \eta_t(1-\kappa)\mathbb{E}\left[\|g_{t,k}\|^2\right]$$

$$+ \eta_t\kappa\frac{\rho_1}{2}\mathbb{E}\left[\|g_{t,k}\|^2\right] + \eta_t\kappa\frac{1}{2\rho_1}\mathbb{E}\left[\|v_e^{t-1}\|^2\right]$$

$$\overset{(b)}{=} \eta_t\frac{\rho_0}{2}L^2\left(\frac{\eta_t\lambda}{1-\lambda}\right)^2\mathbb{E}\left[\|v_{t,k-1}\|^2\right] + \eta_t\frac{1}{\rho_0}(1-\kappa)^2\mathbb{E}\left[\|g_{t,k}\|^2\right] + \eta_t\frac{1}{\rho_0}\kappa^2\mathbb{E}\left[\|v_e^{t-1}\|^2\right] - \eta_t(1-\kappa)\mathbb{E}\left[\|g_{t,k}\|^2\right]$$

$$+ \eta_t\kappa\frac{\rho_1}{2}\mathbb{E}\left[\|g_{t,k}\|^2\right] + \eta_t\kappa\frac{1}{2\rho_1}\mathbb{E}\left[\|v_e^{t-1}\|^2\right],$$

where (a) uses the inequality $\langle a, b\rangle \leq \frac{\rho}{2}\|a\|^2 + \frac{1}{2\rho}\|b\|^2$, and (b) follows from Lemma B.4 where $z_{t,k} = \frac{1}{1-\lambda}\theta_{t,k} - \frac{\lambda}{1-\lambda}\theta_{t,k-1}$, which directly implies $z_{t,k} - \theta_{t,k} = -\frac{\eta_t\lambda}{1-\lambda}v_{t,k-1}$. Therefore:

$$S_0 \leq \eta_t\frac{\rho_0}{2}L^2\left(\frac{\eta_t\lambda}{1-\lambda}\right)^2\left[2\mathbb{E}\|v_{t,k-1} - (1-\lambda)\sum_{i=1}^{k-1}\lambda^{k-1-i}g_{t,i}\|^2 + 2\mathbb{E}\left[\|(1-\lambda)\sum_{i=1}^{k-1}\lambda^{k-1-i}g_{t,i}\|^2\right]\right]$$

$$+ \left[\eta_t\frac{1}{\rho_0}(1-\kappa)^2 - \eta_t(1-\kappa) + \eta_t\kappa\frac{\rho_1}{2}\right]\mathbb{E}\left[\|g_{t,k}\|^2\right] + \left[\eta_t\frac{1}{\rho_0}\kappa^2 + \eta_t\kappa\frac{1}{2\rho_1}\right]\mathbb{E}\left[\|v_e^{t-1}\|^2\right]$$

$$\overset{(c)}{\leq} \eta_t\frac{\rho_0}{2}L^2\left(\frac{\eta_t\lambda}{1-\lambda}\right)^2\left[2\frac{1-\lambda}{1+\lambda}\left[1 - \lambda^{2(k-1)}\right]\frac{\sigma^2}{b} + 2\mathbb{E}\left[\|(1-\lambda)\sum_{i=1}^{k-1}\lambda^{k-1-i}g_{t,i}\|^2\right]\right]$$

$$+ \left[\eta_t\frac{1}{\rho_0}(1-\kappa)^2 - \eta_t(1-\kappa) + \eta_t\kappa\frac{\rho_1}{2}\right]\mathbb{E}\left[\|g_{t,k}\|^2\right] + \left[\eta_t\frac{1}{\rho_0}\kappa^2 + \eta_t\kappa\frac{1}{2\rho_1}\right]\mathbb{E}\left[\|v_e^{t-1}\|^2\right]$$

$$\leq \eta_t\rho_0\left(\frac{\eta_t\lambda L}{1-\lambda}\right)^2\left[\frac{1-\lambda}{1+\lambda}\left[1 - \lambda^{2(k-1)}\right]\frac{\sigma^2}{b} + 2\left(1 - \lambda^{k-1}\right)^2\left[\mathbb{E}\left[\|g_{t,k}\|^2\right] + \mathbb{E}\left[\|\frac{1-\lambda}{1-\lambda^{k-1}}\sum_{i=1}^{k-1}\lambda^{k-1-i}g_{t,i} - g_{t,k}\|^2\right]\right]\right]$$

$$+ \left[\eta_t\frac{1}{\rho_0}(1-\kappa)^2 - \eta_t(1-\kappa) + \eta_t\kappa\frac{\rho_1}{2}\right]\mathbb{E}\left[\|g_{t,k}\|^2\right] + \left[\eta_t\frac{1}{\rho_0}\kappa^2 + \eta_t\kappa\frac{1}{2\rho_1}\right]\mathbb{E}\left[\|v_e^{t-1}\|^2\right]$$

$$\overset{(d)}{\leq} \left[\eta_t\frac{1}{\rho_0}(1-\kappa)^2 - \eta_t(1-\kappa) + \eta_t\kappa\frac{\rho_1}{2} + 2\left(1 - \lambda^{k-1}\right)^2\eta_t\rho_0L^2\left(\frac{\eta_t\lambda}{1-\lambda}\right)^2\right]\mathbb{E}\left[\|g_{t,k}\|^2\right]$$

$$+ \left[\eta_t\frac{1}{\rho_0}\kappa^2 + \eta_t\kappa\frac{1}{2\rho_1}\right]\mathbb{E}\left[\|v_e^{t-1}\|^2\right]$$

$$+ \eta_t\rho_0L^2\left(\frac{\eta_t\lambda}{1-\lambda}\right)^2\frac{1-\lambda}{1+\lambda}\left[1 - \lambda^{2(k-1)}\right]\frac{\sigma^2}{b} + 2\left(1 - \lambda^k\right)^2\eta_t\rho_0L^2\left(\frac{\eta_t}{1-\lambda}\right)^2\mathbb{E}\left[\|\frac{1-\lambda}{1-\lambda^k}\sum_{i=1}^{k}\lambda^{k-i}g_{t,i} - g_{t,k}\|^2\right],$$

$$\tag{14}$$

where (c) employs Lemma B.2, and (d) utilizes the following relation from Lemma B.3:

$$\mathbb{E}\left[\|\frac{1-\lambda}{1-\lambda^k}\sum_{i=1}^{k}\lambda^{k-i}g_{t,i} - g_{t,k}\|^2\right] = \lambda^2\left(\frac{1-\lambda^{k-1}}{1-\lambda^k}\right)\mathbb{E}\left[\|\frac{1-\lambda}{1-\lambda^{k-1}}\sum_{i=1}^{k-1}\lambda^{k-1-i}g_{t,i} - g_{t,k}\|^2\right].$$

We proceed to estimate the upper bound of $S_1$:

$$
\begin{aligned}
S_1 &= \frac{L\eta_t^2}{2} \mathbb{E}\left[\left\|(1-\kappa)\,\tilde{g}_{t,k} + \kappa v_e^{t-1}\right\|^2\right] \\
&\leq L\eta_t^2 \left(1-\kappa\right)^2 \mathbb{E}\left[\|\tilde{g}_{t,k}\|\right] + L\eta_t^2 \kappa^2 \mathbb{E}\left[\|v_e^{t-1}\|\right] \qquad\qquad (15) \\
&\stackrel{(e)}{\leq} L\eta_t^2 \left(1-\kappa\right)^2 \mathbb{E}\left[\|g_{t,k}\|\right] + L\eta_t^2 \left(1-\kappa\right)^2 \frac{\sigma^2}{b} + L\eta_t^2 \kappa^2 \mathbb{E}\left[\|v_e^{t-1}\|\right],
\end{aligned}
$$

where inequality (e) follows from Lemma B.1. Combining equation 14 and equation 15 yields:

$$
\begin{aligned}
\mathbb{E}\left[f\left(z_{t,k+1}\right)\right] &\stackrel{(e)}{\leq} \mathbb{E}\left[f\left(z_{t,k}\right)\right] + \left[\eta_t \frac{1}{\rho_0} \kappa^2 + \eta_t \kappa \frac{1}{2\rho_1} + L\eta_t^2 \kappa^2\right] \mathbb{E}\left[\|v_e^{t-1}\|^2\right] \\
&\quad + \left[\eta_t \frac{1}{\rho_0}\left(1-\kappa\right)^2 - \eta_t\left(1-\kappa\right) + \eta_t \kappa \frac{\rho_1}{2} + 2\eta_t \rho_0 L^2 \left(\frac{\eta_t \lambda}{1-\lambda}\right)^2 + L\eta_t^2 \left(1-\kappa\right)^2\right] \mathbb{E}\left[\|g_{t,k}\|^2\right] \\
&\quad + \left[\eta_t \rho_0 L^2 \left(\frac{\eta_t \lambda}{1-\lambda}\right)^2 \frac{1-\lambda}{1+\lambda} + L\eta_t^2 \left(1-\kappa\right)^2\right] \frac{\sigma^2}{b} \\
&\quad + 2\left(1-\lambda^k\right)^2 \eta_t^3 \rho_0 L^2 \left(\frac{1}{1-\lambda}\right)^2 \mathbb{E}\left[\|\frac{1-\lambda}{1-\lambda^k}\sum_{i=1}^{k}\lambda^{k-i}g_{t,i} - g_{t,k}\|^2\right],
\end{aligned}
$$

Inequality (e) holds since $1 - \lambda^{2(k-1)} \leq 1$ and $1 - \lambda^{k-1} \leq 1$. Recalling the definition $L_k^t = f(z_{t,k}) - f^* + \sum_{i=1}^{k-1} c_i \|\theta_{t,k+1-i} - \theta_{t,k-i}\|^2$, we further obtain:

$$
\begin{aligned}
\mathbb{E}\left[L_{k+1}^t - L_k^t\right] &\leq \left[\eta_t \frac{1}{\rho_0}\kappa^2 + \eta_t \kappa \frac{1}{2\rho_1} + L\eta_t^2 \kappa^2\right] \mathbb{E}\left[\|v_e^{t-1}\|^2\right] + c_1 \mathbb{E}\left[\|\theta_{t,k+1} - \theta_{t,k}\|^2\right] \\
&\quad + \left[\eta_t \frac{1}{\rho_0}\left(1-\kappa\right)^2 - \eta_t\left(1-\kappa\right) + \eta_t \kappa \frac{\rho_1}{2} + 2\eta_t \rho_0 L^2 \left(\frac{\eta_t \lambda}{1-\lambda}\right)^2 + L\eta_t^2 \left(1-\kappa\right)^2\right] \mathbb{E}\left[\|g_{t,k}\|^2\right] \\
&\quad + \left[\eta_t \rho_0 L^2 \left(\frac{\eta_t \lambda}{1-\lambda}\right)^2 \frac{1-\lambda}{1+\lambda} + L\eta_t^2 \left(1-\kappa\right)^2\right] \frac{\sigma^2}{b} + \sum_{i=1}^{k-1}\left(c_{i+1} - c_i\right) \mathbb{E}\left[\|\theta_{t,k+1-i} - \theta_{t,k-i}\|^2\right] \\
&\quad + 2\left(1-\lambda^k\right)^2 \eta_t^3 \rho_0 L^2 \left(\frac{1}{1-\lambda}\right)^2 \mathbb{E}\left[\|\frac{1-\lambda}{1-\lambda^k}\sum_{i=1}^{k}\lambda^{k-i}g_{t,i} - g_{t,k}\|^2\right]
\end{aligned}
$$

From Algorithm 2, we have:

$$
\begin{aligned}
c_1 \mathbb{E}\left[\|\theta_{t,k+1} - \theta_{t,k}\|^2\right] &= c_1 \eta_t^2 \mathbb{E}\left[\|\kappa v_e^{t-1} + (1-\kappa)v_{t,k}\|^2\right] \\
&\leq 2c_1\left(\eta_t \kappa\right)^2 \mathbb{E}\left[\|v_e^{t-1}\|^2\right] + 2c_1\eta_t^2\left(1-\kappa\right)^2 \mathbb{E}\left[\|v_e^{t-1}\|^2\right] \\
&\leq 2c_1\left(\eta_t \kappa\right)^2 \mathbb{E}\left[\|v_e^{t-1}\|^2\right] + 4c_1\eta_t^2\left(1-\kappa\right)^2 \frac{1-\lambda}{1+\lambda}\frac{\sigma^2}{b} + 4c_1\eta_t^2\left(1-\kappa\right)^2 \mathbb{E}\left[\|(1-\lambda)\sum_{i=1}^{k}\lambda^{k-i}g_{t,i}\|^2\right] \\
&\leq 2c_1\left(\eta_t \kappa\right)^2 \mathbb{E}\left[\|v_e^{t-1}\|^2\right] + 4c_1\eta_t^2\left(1-\kappa\right)^2 \frac{1-\lambda}{1+\lambda}\frac{\sigma^2}{b} + 8c_1\eta_t^2\left(1-\kappa\right)^2\left(1-\lambda^k\right)^2 \mathbb{E}\left[\|g_{t,k}\|^2\right] \\
&\quad + 8c_1\eta_t^2\left(1-\kappa\right)^2\left(1-\lambda^k\right)^2 \mathbb{E}\left[\|\frac{1-\lambda}{1-\lambda^k}\sum_{i=1}^{k}\lambda^{k-i}g_{t,i} - g_{t,k}\|^2\right].
\end{aligned}
$$

Thus, we obtain:

$$\mathbb{E}\left[L_{k+1}^t - L_k^t\right] \leq \left[\eta_t \frac{1}{\rho_0}\kappa^2 + \eta_t\kappa\frac{1}{2\rho_1} + L\eta_t^2\kappa^2 + 2c_1\left(\eta_t\kappa\right)^2\right]\mathbb{E}\left[\|v_e^{t-1}\|^2\right]$$

$$+ \left[\eta_t\frac{1}{\rho_0}\left(1-\kappa\right)^2 - \eta_t\left(1-\kappa\right) + \eta_t\kappa\frac{\rho_1}{2} + 2\eta_t\rho_0 L^2\left(\frac{\eta_t\lambda}{1-\lambda}\right)^2 + \left(L\eta_t^2 + 8c_1\eta_t^2\right)\left(1-\kappa\right)^2\right]\mathbb{E}\left[\|g_{t,k}\|^2\right]$$

$$+ \left[\eta_t\rho_0 L^2\left(\frac{\eta_t\lambda}{1-\lambda}\right)^2\frac{1-\lambda}{1+\lambda} + \left(L\eta_t^2 + 4c_1\eta_t^2\frac{1-\lambda}{1+\lambda}\right)\left(1-\kappa\right)^2\right]\frac{\sigma^2}{b}$$

$$+ \sum_{i=1}^{k-1}\left(c_{i+1} - c_i\right)\mathbb{E}\left[\|\theta_{t,k+1-i} - \theta_{t,k-i}\|^2\right]$$

$$+ \left[2\left(1-\lambda^k\right)^2\eta_t^3\rho_0 L^2\left(\frac{1}{1-\lambda}\right)^2 + 8c_1\eta_t^2\left(1-\kappa\right)^2\left(1-\lambda^k\right)^2\right]\mathbb{E}\left[\|\frac{1-\lambda}{1-\lambda^k}\sum_{i=1}^k\lambda^{k-i}g_{t,i} - g_{t,k}\|^2\right].$$

Lemma B.4 can be equivalently expressed as:

$$\mathbb{E}\left[\|\frac{1-\lambda}{1-\lambda^k}\sum_{i=1}^k\lambda^{k-i}g_{t,i} - g_{t,k}\|^2\right] \leq \sum_{i=1}^{k-1}a_{k,k-i}\mathbb{E}\left[\|\theta_{t,k+1-i} - \theta_{t,k-i}\|^2\right],$$

where $a_{k,k-i} = \frac{L^2\lambda^i}{1-\lambda^k}\left(i + \frac{\lambda}{1-\lambda}\right)$. Substituting this into the expression for $\mathbb{E}\left[L_{k+1}^t - L_k^t\right]$ gives:

$$\mathbb{E}\left[L_{k+1}^t - L_k^t\right] \leq \left[\eta_t\frac{1}{\rho_0}\kappa^2 + \eta_t\kappa\frac{1}{2\rho_1} + L\eta_t^2\kappa^2 + 2c_1\left(\eta_t\kappa\right)^2\right]\mathbb{E}\left[\|v_e^{t-1}\|^2\right]$$

$$+ \left[\eta_t\frac{1}{\rho_0}\left(1-\kappa\right)^2 - \eta_t\left(1-\kappa\right) + \eta_t\kappa\frac{\rho_1}{2} + 2\left(1-\lambda^{k-1}\right)^2\eta_t\rho_0 L^2\left(\frac{\eta_t\lambda}{1-\lambda}\right)^2 + \left(L\eta_t^2 + 8c_1\eta_t^2\right)\left(1-\kappa\right)^2\right]$$

$$\cdot \mathbb{E}\left[\|g_{t,k}\|^2\right]$$

$$+ \left[\eta_t\rho_0 L^2\left(\frac{\eta_t\lambda}{1-\lambda}\right)^2\frac{1-\lambda}{1+\lambda} + \left(L\eta_t^2 + 4c_1\eta_t^2\frac{1-\lambda}{1+\lambda}\right)\left(1-\kappa\right)^2\right]\frac{\sigma^2}{b}$$

$$+ \sum_{i=1}^{k-1}\left(c_{i+1} - c_i\right)\mathbb{E}\left[\|\theta_{t,k+1-i} - \theta_{t,k-i}\|^2\right]$$

$$+ \left[2\left(1-\lambda^k\right)^2\eta_t^3\rho_0 L^2\left(\frac{1}{1-\lambda}\right)^2 + 8c_1\eta_t^2\left(1-\kappa\right)^2\left(1-\lambda^k\right)^2\right]\sum_{i=1}^{k-1}a_{k,k-i}\mathbb{E}\left[\|\theta_{t,k+1-i} - \theta_{t,k-i}\|^2\right],$$

To ensure the sum of the coefficients of the last two terms is non-positive, the following condition must hold for all $i \geq 1$:

$$c_i + 1 \leq c_i - \left[2\left(1-\lambda^k\right)^2\eta_t^3\rho_0 L^2\left(\frac{1}{1-\lambda}\right)^2 + 8\left(1-\lambda^k\right)^2 c_1\eta_t^2\left(1-\kappa\right)^2\right]a_{k,k-i},$$

Given that $\left(1-\lambda^k\right)^2 < 1$ always holds, it suffices to satisfy the following inequality for all $i \geq 1$:

$$c_i + 1 = c_i - \left[2\eta_t^3\rho_0 L^2\left(\frac{1}{1-\lambda}\right)^2 + 8c_1\eta_t^2\left(1-\kappa\right)^2\right]a_{k,k-i}.$$

Moreover, to guarantee $c_i \geq 0$ for all $i \geq 1$, we can define $c_i$ as:

$$c_1 = \left[2\eta_t^3\rho_0 L^2\left(\frac{1}{1-\lambda}\right)^2 + 8c_1\eta_t^2\left(1-\kappa\right)^2\right]L^2\sum_{i=1}^{\infty}\lambda^i\left(i + \frac{\lambda}{1-\lambda}\right),$$

This yields:

$$c_1 = \frac{2\eta_t^3\rho_0 L^4\frac{\lambda+\lambda^2}{(1-\lambda)^4}}{1 - 8\eta_t^2 L^2\left(1-\kappa\right)^2\frac{\lambda+\lambda^2}{(1-\lambda)^2}}. \tag{16}$$

Furthermore, setting $\eta_t \leq \frac{1-\lambda}{2\sqrt{2}L(1-\kappa)\sqrt{\lambda+\lambda^2}}$ ensures $c_1 \geq 0$. Consequently, $\mathbb{E}\left[L_{k+1}^t - L_k^t\right]$ simplifies to:

$$
\mathbb{E}\left[L_{k+1}^t - L_k^t\right] \leq \left[\eta_t \frac{1}{\rho_0}\kappa^2 + \eta_t \kappa \frac{1}{2\rho_1} + L\eta_t^2\kappa^2 + 2c_1\left(\eta_t\kappa\right)^2\right] \mathbb{E}\left[\|v_e^{t-1}\|^2\right]
$$

$$
+ \left[\eta_t \frac{1}{\rho_0}\left(1-\kappa\right)^2 - \eta_t\left(1-\kappa\right) + \eta_t\kappa\frac{\rho_1}{2} + 2\eta_t\rho_0 L^2\left(\frac{\eta_t\lambda}{1-\lambda}\right)^2 + \left(L\eta_t^2 + 8c_1\eta_t^2\right)\left(1-\kappa\right)^2\right] \mathbb{E}\left[\|g_{t,k}\|^2\right]
$$

$$
+ \left[\eta_t\rho_0 L^2\left(\frac{\eta_t\lambda}{1-\lambda}\right)^2\frac{1-\lambda}{1+\lambda} + \left(L\eta_t^2 + 4c_1\eta_t^2\frac{1-\lambda}{1+\lambda}\right)\left(1-\kappa\right)^2\right]\frac{\sigma^2}{b}.
$$

This can be rewritten as:

$$
\mathbb{E}\left[L_{k+1}^t - L_k^t\right] \leq -M_1\mathbb{E}\left[\|g_{t,k}\|^2\right] + M_2\mathbb{E}\left[\|v_e^{t-1}\|^2\right] + M_3, \tag{17}
$$

where:

$$
M_1 = \eta_t\left(1-\kappa\right) - \eta_t\frac{1}{\rho_0}\left(1-\kappa\right)^2 - \eta_t\kappa\frac{\rho_1}{2} - 2\eta_t\rho_0 L^2\left(\frac{\eta_t\lambda}{1-\lambda}\right)^2 - \left(L\eta_t^2 + 8c_1\eta_t^2\right)\left(1-\kappa\right)^2,
$$
$$
\tag{18}
$$

$$
M_2 = \eta_t\frac{1}{\rho_0}\kappa^2 + \eta_t\kappa\frac{1}{2\rho_1} + L\eta_t^2\kappa^2 + 2c_1\left(\eta_t\kappa\right)^2, \tag{19}
$$

$$
M_3 = \left[\eta_t\rho_0 L^2\left(\frac{\eta_t\lambda}{1-\lambda}\right)^2\frac{1-\lambda}{1+\lambda} + \left(L\eta_t^2 + 4c_1\eta_t^2\frac{1-\lambda}{1+\lambda}\right)\left(1-\kappa\right)^2\right]\frac{\sigma^2}{b}. \tag{20}
$$

Summing equation 16 from $i=1$ to $k+1$ gives:

$$
L_1^t \geq \mathbb{E}\left[L_1^t - L_{k+1}^t\right] \geq M_1\sum_{i=1}^{k}\mathbb{E}\left[\|g_{t,i}\|^2\right] - kM_2\mathbb{E}\left[\|v_e^{t-1}\|^2\right] - kM_3. \tag{21}
$$

From equation 20, it immediately follows that:

$$
\sum_{i=1}^{\lfloor\frac{B}{b}\rfloor}\mathbb{E}\left[\|g_{t,i}\|^2\right] \leq \frac{L_1^t}{M_1} + \frac{B}{b}\left[\frac{M_2}{M_1}\mathbb{E}\left[\|v_e^{t-1}\|^2\right] + \frac{M_3}{M_1}\right], \tag{22}
$$

$$
\sum_{i=1}^{\lfloor\frac{B}{b}\rfloor-1}\mathbb{E}\left[\|g_{t,i}\|^2\right] \leq \frac{L_1^t}{M_1} + \left(\frac{B}{b}-1\right)\left[\frac{M_2}{M_1}\mathbb{E}\left[\|v_e^{t-1}\|^2\right] + \frac{M_3}{M_1}\right], \tag{23}
$$

Subtracting equation 22 from equation 21 yields:

$$
\mathbb{E}\left[\|g_{t,\lfloor\frac{B}{b}\rfloor}\|^2\right] = \mathbb{E}\left[\|g_t\|^2\right] \leq \frac{M_2}{M_1}\mathbb{E}\left[\|v_e^{t-1}\|^2\right] + \frac{M_3}{M_1}. \tag{24}
$$

Finally, summing equation 23 from $t=0$ to $T-1$ results in:

$$
\frac{1}{T}\sum_{t=0}^{T-1}\mathbb{E}\left[\|g_t\|^2\right] \leq \frac{M_3}{M_1} + \frac{M_2}{M_1}\frac{1}{T}\sum_{t=0}^{T-1}\mathbb{E}\left[\|v_e^{t-1}\|^2\right]. \tag{25}
$$

We now appropriately bound the terms $M_1$, $M_2$, and $M_3$ to simplify the final result. Setting $\eta_t \leq \frac{1-\lambda}{4L(1-\kappa)\sqrt{\lambda+\lambda^2}}$ implies:

$$
1 - 8\eta_t^2 L^2\left(1-\kappa\right)^2\frac{\lambda+\lambda^2}{\left(1-\lambda\right)^2} \geq \frac{1}{2}.
$$

Additionally, the condition $\eta_t \leq \frac{1-\lambda}{2\sqrt{2}L(1-\kappa)\sqrt{\lambda+\lambda^2}}$ directly gives:

$$c_1 \leq 4\eta_t^3 \rho_0 L^4 \frac{\lambda + \lambda^2}{(1-\lambda)^4} \leq \frac{1}{2}\eta_t \rho_0 \frac{L^2}{(1-\lambda)^2 (1-\kappa)^2}. \tag{26}$$

Now, define $\rho_0 = \frac{(1-\lambda)(1-\kappa)}{2L\eta_t}$ and $\rho_1 = \frac{1-\kappa}{2\kappa}$. According to equation 18, we have:

$$
\begin{aligned}
M_1 &= \eta_t (1-\kappa) - \eta_t \frac{1}{\rho_0} (1-\kappa)^2 - \eta_t \kappa \frac{\rho_1}{2} - 2\eta_t \rho_0 L^2 \left(\frac{\eta_t \lambda}{1-\lambda}\right)^2 - \left(L\eta_t^2 + 8c_1\eta_t^2\right)(1-\kappa)^2 \\
&\overset{(f)}{\geq} \eta_t (1-\kappa) - \eta_t \frac{1}{\rho_0} (1-\kappa)^2 - \eta_t \kappa \frac{\rho_1}{2} - 2\eta_t \rho_0 L^2 \left(\frac{\eta_t \lambda}{1-\lambda}\right)^2 - \left(L\eta_t^2 + 4\eta_t^3 \rho_0 \frac{L^2}{(1-\lambda)^2 (1-\kappa)^2}\right)(1-\kappa)^2 \\
&\overset{(g)}{=} \eta_t (1-\kappa) - \frac{2L\eta_t^2 (1-\kappa)}{(1-\lambda)} - \frac{\eta_t (1-\kappa)}{2} - \frac{L\eta_t^2 \lambda^2 (1-\kappa)}{1-\lambda} - L\eta_t^2 (1-\kappa)^2 - \frac{2L\eta_t^2 (1-\kappa)}{1-\lambda} \\
&\geq \frac{1}{2}\eta_t (1-\kappa) - L\eta_t^2 \left[\frac{4(1-\kappa)}{1-\lambda} + (1-\kappa)^2 + \frac{\lambda^2 (1-\kappa)}{1-\lambda}\right] \\
&= \frac{1}{2}\eta_t (1-\kappa) - L\eta_t^2 \frac{\left(4+\lambda^2\right)(1-\kappa) + (1-\lambda)(1-\kappa)^2}{1-\lambda} \\
&\overset{(h)}{\geq} \frac{\eta_t (1-\kappa)}{4},
\end{aligned}
$$

where (f) utilizes equation 26, (g) substitutes the definitions of $\rho_0$ and $\rho_1$, and (h) holds under the condition $\eta_t \leq \frac{1-\lambda}{4L[(4+\lambda^2)+(1-\lambda)(1-\kappa)]}$. This leads to:

$$M_1 \geq \frac{\eta_t (1-\kappa)}{4}. \tag{27}$$

Subsequently, for $M_2$ and $M_3$, combining equation 19, equation 20, $\rho_0 = \frac{(1-\lambda)(1-\kappa)}{2L\eta_t}$, and $\rho_1 = \frac{1-\kappa}{2\kappa}$ yields:

$$
\begin{aligned}
M_2 &= \eta_t \frac{1}{\rho_0}\kappa^2 + \eta_t \kappa \frac{1}{2\rho_1} + L\eta_t^2 \kappa^2 + 2c_1 (\eta_t \kappa)^2 \\
&\overset{(i)}{\leq} \frac{2L\eta_t^2 \kappa^2}{(1-\lambda)(1-\kappa)} + \frac{\kappa^2 \eta_t}{1-\kappa} + L\eta_t^2 \kappa^2 + \frac{L\eta_t^2 \kappa^2}{2(1-\lambda)(1-\kappa)} \\
&= \frac{\kappa^2 \eta_t}{1-\kappa} + L\eta_t^2 \left[\frac{2\kappa^2}{(1-\lambda)(1-\kappa)} + \kappa^2 + \frac{\kappa^2}{2(1-\lambda)(1-\kappa)}\right] \\
&= \frac{\kappa^2 \eta_t}{1-\kappa} + L\eta_t \frac{[5 + 2(1-\lambda)(1-\kappa)]\kappa^2}{2(1-\lambda)(1-\kappa)} \cdot \frac{1-\lambda}{4L[(4+\lambda^2)+(1-\lambda)(1-\kappa)]} \\
&= \frac{\kappa^2 \eta_t}{1-\kappa} + \eta_t \frac{[5 + 2(1-\lambda)(1-\kappa)]\kappa^2}{8(1-\kappa)[(4+\lambda^2)+(1-\lambda)(1-\kappa)]} \\
&\overset{(j)}{\leq} \frac{2\kappa^2 \eta_t}{1-\kappa},
\end{aligned}
$$

where (i) similarly utilizes equation 26 and substitutes $\rho_0$ and $\rho_1$, (j) holds under the condition $\eta_t \leq \frac{1-\lambda}{4L[(4+\lambda^2)+(1-\lambda)(1-\kappa)]}$, and the final inequality follows because $\frac{[5+2(1-\lambda)(1-\kappa)]\kappa^2}{8(1-\kappa)[(4+\lambda^2)+(1-\lambda)(1-\kappa)]} \leq \frac{\kappa^2}{1-\kappa}$ evidently holds. Consequently, we obtain:

$$M_2 \leq \frac{2\kappa^2 \eta_t}{1-\kappa}. \tag{28}$$

For $M_3$, we have:

$$M_3 = \left[ \eta_t \rho_0 L^2 \left( \frac{\eta_t \lambda}{1-\lambda} \right)^2 \frac{1-\lambda}{1+\lambda} + \left( L\eta_t^2 + 4c_1\eta_t^2 \frac{1-\lambda}{1+\lambda} \right) (1-\kappa)^2 \right] \frac{\sigma^2}{b}$$

$$\leq \left[ \eta_t \rho_0 L^2 \left( \frac{\eta_t \lambda}{1-\lambda} \right)^2 \frac{1-\lambda}{1+\lambda} + L\eta_t^2 (1-\kappa)^2 + 16\eta_t^3 \rho_0 L^4 \frac{\lambda + \lambda^2}{(1-\lambda)^4} \frac{1-\lambda}{1+\lambda} \eta_t^2 (1-\kappa)^2 \right] \frac{\sigma^2}{b}$$

$$= \left[ \frac{L\eta_t^2 \lambda^2 (1-\kappa)}{2(1+\lambda)} + L\eta_t^2 (1-\kappa)^2 + \frac{8\lambda}{(1-\lambda)^2} L^3 \eta_t^4 (1-\kappa)^3 \right] \frac{\sigma^2}{b}$$

$$\overset{(e)}{\leq} \left[ \frac{L\eta_t^2 \lambda^2 (1-\kappa)}{2(1+\lambda)} + L\eta_t^2 (1-\kappa)^2 + \frac{\lambda}{2} L\eta_t^2 (1-\kappa) \right] \frac{\sigma^2}{b},$$

where (e) holds since $\eta_t \leq \min\left\{ \frac{1-\lambda}{4L[(4+\lambda^2)+2(1-\lambda)(1-\kappa)]}, \frac{1-\lambda}{2\sqrt{2}L(1-\kappa)\sqrt{\lambda+\lambda^2}} \right\} \leq \frac{1-\lambda}{4L(1-\kappa)}$. Therefore:

$$M_3 \leq \left[ \frac{L\eta_t^2 \lambda^2 (1-\kappa)}{2(1+\lambda)} + L\eta_t^2 (1-\kappa)^2 + \frac{\lambda}{2} L\eta_t^2 (1-\kappa) \right] \frac{\sigma^2}{b}, \tag{29}$$

Finally, an appropriate bounding for $\mathbb{E}\left[ \|v_e^{t-1}\|^2 \right]$ is required. From Algorithm 2 and Lemma B.1, we derive:

$$\mathbb{E}\left[ \|v_e^{t-1}\|^2 \right] = \mathbb{E}\left[ \| (1-\lambda)(1-\kappa) \sum_{i=0}^{t-1} \sum_{j=1}^{\lfloor \frac{\mathcal{B}}{b} \rfloor} \kappa^{t-1-i} \lambda^{\lfloor \frac{\mathbb{B}}{b} \rfloor - j} \tilde{g}_j^i \|^2 \right]$$

$$= \mathbb{E}\left[ \| (1-\lambda)(1-\kappa) \sum_{i=0}^{t-1} \sum_{j=1}^{\lfloor \frac{\mathcal{B}}{b} \rfloor} \kappa^{t-1-i} \lambda^{\lfloor \frac{\mathbb{B}}{b} \rfloor - j} \left( g_j^i + \delta_j^i \right) \|^2 \right]$$

$$= \mathbb{E}\left[ \| (1-\lambda)(1-\kappa) \sum_{i=0}^{t-1} \sum_{j=1}^{\lfloor \frac{\mathcal{B}}{b} \rfloor} \kappa^{t-1-i} \lambda^{\lfloor \frac{\mathbb{B}}{b} \rfloor - j} g_j^i \|^2 \right] + \mathbb{E}\left[ \| (1-\lambda)(1-\kappa) \sum_{i=0}^{t-1} \sum_{j=1}^{\lfloor \frac{\mathcal{B}}{b} \rfloor} \kappa^{t-1-i} \lambda^{\lfloor \frac{\mathbb{B}}{b} \rfloor - j} \delta_j^i \|^2 \right]$$

$$\leq \mathbb{E}\left[ \| (1-\lambda)(1-\kappa) \sum_{i=0}^{t-1} \sum_{j=1}^{\lfloor \frac{\mathcal{B}}{b} \rfloor} \kappa^{t-1-i} \lambda^{\lfloor \frac{\mathbb{B}}{b} \rfloor - j} g_j^i \|^2 \right] + (1-\lambda)^2 (1-\kappa)^2 \sum_{i=0}^{t-1} \sum_{j=1}^{\lfloor \frac{\mathcal{B}}{b} \rfloor} \kappa^{2(t-1-i)} \lambda^{2\left( \lfloor \frac{\mathbb{B}}{b} \rfloor - j \right)} \frac{\sigma^2}{b}$$

$$\leq (1-\lambda)(1-\kappa) \sum_{i=0}^{t-1} \sum_{j=1}^{\lfloor \frac{\mathcal{B}}{b} \rfloor} \kappa^{t-1-i} \lambda^{\lfloor \frac{\mathbb{B}}{b} \rfloor - j} \mathbb{E}\left[ \|g_j^i\|^2 \right] + (1-\lambda)^2 (1-\kappa)^2 \sum_{i=0}^{t-1} \sum_{j=1}^{\lfloor \frac{\mathcal{B}}{b} \rfloor} \kappa^{2(t-1-i)} \lambda^{2\left( \lfloor \frac{\mathbb{B}}{b} \rfloor - j \right)} \frac{\sigma^2}{b},$$

Define $c_{i,j} = (1-\lambda)(1-\kappa) \kappa^{t-1-i} \lambda^{\lfloor \frac{\mathcal{B}}{b} \rfloor - j}$. This allows us to express the above as:

$$\mathbb{E}\left[ \|v_e^{t-1}\|^2 \right] \leq \sum_{i=0}^{t-1} \sum_{j=1}^{\lfloor \frac{\mathcal{B}}{b} \rfloor} c_{i,j} \mathbb{E}\left[ \|g_j^i\|^2 \right] + \sum_{i=0}^{t-1} \sum_{j=1}^{\lfloor \frac{\mathcal{B}}{b} \rfloor} c_{i,j}^2 \frac{\sigma^2}{b},$$

Given that $\sum_{i=0}^{t-1} \kappa^{t-1-i} = \frac{1-\kappa^t}{1-\kappa}$ and $\sum_{j=1}^{\lfloor \frac{\mathcal{B}}{b} \rfloor} \lambda^{\lfloor \frac{\mathcal{B}}{b} \rfloor - j} = \frac{1-\lambda^{\lfloor \frac{\mathcal{B}}{b} \rfloor}}{1-\lambda}$, the coefficient sums satisfy $\sum_{i=0}^{t-1} \sum_{j=1}^{\lfloor \frac{\mathcal{B}}{b} \rfloor} c_{i,j} = (1-\kappa^t)\left( 1 - \lambda^{\lfloor \frac{\mathcal{B}}{b} \rfloor} \right) \leq 1$ and $\sum_{i=0}^{t-1} \sum_{j=1}^{\lfloor \frac{\mathcal{B}}{b} \rfloor} c_{i,j}^2 = (1-\lambda)^2 (1-\kappa)^2 \frac{1-\kappa^{2t}}{1-\kappa^2} \frac{1-\lambda^{2\lfloor \frac{\mathcal{B}}{b} \rfloor}}{1-\lambda^2}$. Applying equation 22 yields:

$$\mathbb{E}\left[ \|v_e^{t-1}\|^2 \right] \leq (1-\lambda)(1-\kappa) \sum_{i=0}^{t-1} \kappa^{t-1-i} \left[ \frac{L_1^i}{M_1} + \frac{B}{b} \left[ \frac{M_2}{M_1} \mathbb{E}\left[ \|v_e^{t-1}\|^2 \right] + \frac{M_3}{M_1} \right] \right] + (1-\lambda)^2 (1-\kappa)^2 \frac{1-\kappa^{2t}}{1-\kappa^2} \frac{1-\lambda^{2\lfloor \frac{\mathcal{B}}{b} \rfloor}}{1-\lambda^2} \frac{\sigma^2}{b} \tag{30}$$

Now, let $a_t = \mathbb{E}\left[ \|v_e^{t-1}\|^2 \right]$ with $a_0 = \mathbb{E}\left[ \|v_e^{-1}\|^2 \right] = 0$, and $S_t = \sum_{i=0}^{t} a_i$ with $S_0 = a_0 = 0$. The expression above becomes:

$$a_t \leq (1-\lambda)(1-\kappa) \sum_{i=0}^{t-1} \kappa^{t-1-i} \left[ \frac{L_1^i}{M_1} + \frac{B}{b} \left[ \frac{M_2}{M_1} a_i + \frac{M_3}{M_1} \right] \right] + (1-\lambda)^2 (1-\kappa)^2 \frac{1-\kappa^{2t}}{1-\kappa^2} \frac{1-\lambda^{2\lfloor \frac{\mathcal{B}}{b} \rfloor}}{1-\lambda^2} \frac{\sigma^2}{b}. \tag{31}$$

Summing equation 31 from $t = 0$ to $T - 1$ gives:

$$\sum_{t=0}^{T-1} a_t \leq \sum_{t=0}^{T-1} \left[ (1-\lambda)(1-\kappa) \sum_{i=0}^{t-1} \kappa^{t-1-i} \left[ \frac{L_1^t}{M_1} + \frac{B}{b} \left[ \frac{M_2}{M_1} a_i + \frac{M_3}{M_1} \right] \right] + (1-\lambda)^2 (1-\kappa)^2 \frac{1-\kappa^{2t}}{1-\kappa^2} \frac{1-\lambda^{2\lfloor \frac{\mathbb{B}}{b} \rfloor}}{1-\lambda^2} \frac{\sigma^2}{b} \right]$$

$$= (1-\lambda)(1-\kappa) \sum_{t=1}^{T-1} \sum_{i=0}^{t-1} \kappa^{t-1-i} \frac{L_1^i}{M_1} + (1-\lambda)(1-\kappa) \lfloor \frac{\mathbb{B}}{b} \rfloor \sum_{t=1}^{T-1} \sum_{i=0}^{t-1} \kappa^{t-1-i} \left( \frac{M_2}{M_1} a_i + \frac{M_3}{M_1} \right)$$

$$+ \sum_{t=1}^{T-1} (1-\lambda)^2 (1-\kappa)^2 \frac{1-\kappa^{2t}}{1-\kappa^2} \frac{1-\lambda^{2\lfloor \frac{\mathbb{B}}{b} \rfloor}}{1-\lambda^2} \frac{\sigma^2}{b}$$

$$= (1-\lambda)(1-\kappa) \sum_{i=0}^{T-2} \sum_{t=i+1}^{T-1} \kappa^{t-1-i} \left( \frac{L_1^i}{M_1} + \frac{B}{b} \frac{M_2}{M_1} a_i \right) + (1-\lambda)(1-\kappa) \lfloor \frac{\mathbb{B}}{b} \rfloor \sum_{t=1}^{T-1} \sum_{i=0}^{t-1} \kappa^{t-1-i} \frac{M_3}{M_1}$$

$$+ \sum_{t=1}^{T-1} (1-\lambda)^2 (1-\kappa)^2 \frac{1-\kappa^{2t}}{1-\kappa^2} \frac{1-\lambda^{2\lfloor \frac{\mathbb{B}}{b} \rfloor}}{1-\lambda^2} \frac{\sigma^2}{b}$$

$$\leq (1-\lambda)(1-\kappa) \left( \frac{T-1}{1-\kappa} \frac{L_1^0}{M_1} + \frac{B}{b} \frac{M_2}{M_1} \frac{S_{T-1}}{1-\kappa} \right) + (1-\lambda)(T-1) \lfloor \frac{\mathbb{B}}{b} \rfloor \frac{M_3}{M_1} + \frac{(1-\lambda)(1-\kappa)}{(1+\lambda)(1+\kappa)} (T-1) \frac{\sigma^2}{b}.$$

The last inequality holds because the following condition is satisfied:

$$\sum_{i=0}^{T-2} \sum_{t=i+1}^{T-1} \kappa^{t-1-i} \left( \frac{L_1^i}{M_1} + \frac{B}{b} \frac{M_2}{M_1} a_i \right) \leq \frac{1}{1-\kappa} \sum_{i=0}^{T-2} \left( \frac{L_1^i}{M_1} + \frac{B}{b} \frac{M_2}{M_1} a_i \right) \leq \frac{T-1}{1-\kappa} \frac{L_1^0}{M_1} + \frac{B}{b} \frac{M_2}{M_1} \frac{S_{T-1}}{1-\kappa}, \tag{32}$$

$$\sum_{t=1}^{T-1} \sum_{i=0}^{t-1} \kappa^{t-1-i} = \sum_{t=1}^{T-1} \frac{1-\kappa^t}{1-\kappa} \leq \frac{T-1}{1-\kappa}, \tag{33}$$

$$\sum_{t=1}^{T-1} \sum_{i=0}^{t-1} \kappa^{2(t-1-i)} = \sum_{t=1}^{T-1} \frac{1-\kappa^{2t}}{1-\kappa^2} \leq \frac{T-1}{1-\kappa^2}, \tag{34}$$

$$\sum_{t=1}^{T-1} \frac{1-\lambda^{2\lfloor \frac{\mathbb{B}}{b} \rfloor}}{1-\lambda^2} \leq \sum_{t=1}^{T-1} \frac{1}{1-\lambda^2} = \frac{T-1}{(1-\lambda)(1+\lambda)}. \tag{35}$$

Rearranging yields:

$$S_{T-1} = \sum_{t=0}^{T-1} \mathbb{E}\left[ \|v_e^{t-1}\|^2 \right] \leq \frac{T(1-\lambda)}{1 - \frac{B}{b} \frac{M_2}{M_1} (1-\lambda)} \left( \frac{L_1^0}{M_1} + \lfloor \frac{\mathbb{B}}{b} \rfloor \frac{M_3}{M_1} + \frac{1-\kappa}{(1+\lambda)(1+\kappa)} \frac{\sigma^2}{b} \right). \tag{36}$$

Combining equation 27, equation 28, and equation 29 results in:

$$\frac{M_2}{M_1} \leq 8\kappa^2, \tag{37}$$

$$\frac{M_3}{M_1} \leq \left[ \frac{2L\eta_t \lambda^2}{1+\lambda} + 4L\eta_t (1-\kappa) + 2L\eta_t \lambda \right] \frac{\sigma^2}{b} \leq \frac{8L\eta_t \sigma^2}{b}. \tag{38}$$

Substituting equation 27, equation 28, and equation 29 into equation 36 gives:

$$S_{T-1} \leq \sum_{t=0}^{T-1} \mathbb{E}\left[ \|v_e^{t-1}\|^2 \right] \leq \frac{T(1-\lambda)}{1 - 8\kappa^2 \frac{B}{b} (1-\lambda)} \left( \frac{2L_1^0}{\eta_t} + \lfloor \frac{\mathbb{B}}{b} \rfloor \frac{4L\eta_t \sigma^2}{b} + \frac{1-\kappa}{(1+\lambda)(1+\kappa)} \frac{\sigma^2}{b} \right). \tag{39}$$

Finally, substituting equation 27, equation 28, equation 29, and equation 38 into equation 25 yields the final result, under the condition $\kappa^2 (1-\lambda) \leq \frac{1}{8\frac{B}{b}}$:

$$\frac{1}{T} \sum_{t=0}^{T-1} \mathbb{E}\left[ \|g_t\|^2 \right] \leq \frac{8L\eta_t \sigma^2}{b} + \frac{(1-\lambda)}{1 - 8\kappa^2 \frac{B}{b} (1-\lambda)} \left( \frac{2(f(\theta_0) - f^*)}{\eta_t} + \lfloor \frac{\mathbb{B}}{b} \rfloor \frac{8L\eta_t \sigma^2}{b} + \frac{1-\kappa}{(1+\lambda)(1+\kappa)} \frac{\sigma^2}{b} \right). \tag{40}$$

This completes the proof. $\qquad\square$

## D PROOFS FOR THE CONVERGENCE OF SGD/MOMENTUM WITH MINI-BATCH

*Proof of Theorem 1.* For the convergence analysis of SGDM, we outline the key steps. Starting from Algorithm 1, we derive:

$$
\mathbb{E}\left[\|\theta_{t,k} - \theta_{t,0}\|^2\right] = \mathbb{E}\left[\|\theta_{t,k} - z_{t,k} + z_{t,k} + \theta_{t,0}\|^2\right] \le 2\mathbb{E}\left[\|z_{t,k} - \theta_{t,k}\|^2\right] + 2\mathbb{E}\left[\|z_{t,k} - \theta_{t,0}\|^2\right]
$$

$$
= 2\mathbb{E}\left[\|\frac{1}{1-\lambda}\theta_{t,k} - \frac{\lambda}{1-\lambda}\theta_{t,k-1} - \theta_{t,k}\|^2\right] + 2\mathbb{E}\left[\|z_{t,k} - \theta_{t,0}\|^2\right]
$$

$$
= 2\mathbb{E}\left[\|\frac{\lambda}{1-\lambda}\theta_{t,k} - \frac{\lambda}{1-\lambda}\theta_{t,k-1}\|^2\right] + 2\mathbb{E}\left[\|z_{t,k} - \theta_{t,0}\|^2\right] = 2\frac{\eta_t^2\lambda^2}{(1-\lambda)^2}\mathbb{E}\left[\|v_k^t\|^2\right] + 2\mathbb{E}\left[\|z_{t,k} - \theta_{t,0}\|^2\right]
$$

Regarding $\mathbb{E}\left[\|v_k^t\|^2\right]$, Lemma B.4 yields:

$$
\mathbb{E}\left[\|v_{t,k}\|^2\right] = (1-\lambda)^2\mathbb{E}\left[\|\sum_{i=0}^{k-1}\lambda^{k-1-i}\tilde{g}_{t,k}\|^2\right] = \left(\frac{1-\lambda}{b}\right)^2\mathbb{E}\left[\|\sum_{i=0}^{k-1}\lambda^{k-1-i}\sum_{j=1}^{b}\nabla f(\theta_{t,i};\xi_{i,j})\|^2\right]
$$

$$
\le \left(\frac{1-\lambda}{b}\right)^2\left[\mathbb{E}\left[\|\sum_{i=0}^{k-1}\lambda^{k-1-i}\sum_{j=1}^{b}[\nabla f(\theta_{t,i};\xi_{i,j}) - \nabla f(\theta_{t,0})]\|^2\right] + \mathbb{E}\left[\|\frac{b(1-\lambda^k)}{1-\lambda}\nabla f(\theta_{t,0})\|^2\right]\right]
$$

$$
\le (1-\lambda)^2\frac{2k}{b}\sum_{i=0}^{k-1}\sum_{j=1}^{b}\mathbb{E}\left[\|\nabla f(\theta_{t,i};\xi_{i,j}) - \nabla f(\theta_{t,0})\|^2\right] + 2\mathbb{E}\left[\|\nabla f(\theta_{t,0})\|^2\right]
$$

$$
\overset{(a)}{\le} (1-\lambda)^2\frac{2Bk}{b}\left[\ominus\mathbb{E}\left[\nabla f(\theta_{t,0})\right] + \frac{\sigma^2}{b}\right] + 2\mathbb{E}\left[\|\nabla f(\theta_{t,0})\|^2\right].
$$

Furthermore, we derive:

$$
\mathbb{E}\left[\|z_{t,i} - z_{t,0}\|^2\right] = \mathbb{E}\left[\|\frac{\eta_t}{b}\sum_{k=0}^{i-1}\sum_{j=1}^{b}\nabla f(\theta_{t,k};\xi_{k,j})\|^2\right] = i^2\eta_t^2\mathbb{E}\left[\|\frac{1}{ib}\sum_{k=0}^{i-1}\sum_{j=1}^{b}\nabla f(\theta_{t,k};\xi_{k,j})\|^2\right]
$$

$$
\le 3i^2\eta_t^2\mathbb{E}\left[\|\frac{1}{ib}\sum_{k=0}^{i-1}\sum_{j=1}^{b}[\nabla f(\theta_{t,k};\xi_{k,j}) - \nabla f(\theta_{t,0};\xi_{k,j})]\|^2\right]
$$

$$
+ 3i^2\eta_t^2\mathbb{E}\left[\|\frac{1}{ib}\sum_{k=0}^{i-1}\sum_{j=1}^{b}[\nabla f(\theta_{t,0};\xi_{k,j}) - \nabla f(\theta_{t,0})]\|^2\right] + 3i^2\eta_t^2\mathbb{E}\left[\|\nabla f(\theta_{t,0})\|^2\right]
$$

$$
\overset{(b)}{\le} \frac{3i^2\eta_t^2}{ib}\sum_{k=0}^{i-1}\sum_{j=1}^{b}\mathbb{E}\left[\|[\nabla f(\theta_{t,k};\xi_{k,j}) - \nabla f(\theta_{t,0};\xi_{k,j})]\|^2\right] + 3i\eta_t^2\frac{B}{b}\left[\ominus\mathbb{E}\left[\nabla f(\theta_{t,0})\right] + \frac{\sigma^2}{b}\right]
$$

$$
+ 3i^2\eta_t^2\mathbb{E}\left[\|\nabla f(\theta_{t,0})\|^2\right]
$$

$$
\le \frac{3i^2\eta_t^2}{ib}\sum_{k=0}^{i-1}\sum_{j=1}^{b}L^2\mathbb{E}\left[\|[\theta_{t,k} - \theta_{t,0}]\|^2\right] + 3i\eta_t^2\frac{B}{b}\left[\ominus\mathbb{E}\left[\nabla f(\theta_{t,0})\right] + \frac{\sigma^2}{b}\right] + 3i^2\eta_t^2\mathbb{E}\left[\|\nabla f(\theta_{t,0})\|^2\right]
$$

$$
\le 3i\eta_t^2L^2\sum_{k=0}^{\lfloor\frac{B}{b}\rfloor-1}\mathbb{E}\left[\|\theta_{t,k} - \theta_{t,0}\|^2\right] + 3\eta_t^2\left[i\frac{B}{b}\left[\ominus\mathbb{E}\left[\nabla f(\theta_{t,0})\right] + \frac{\sigma^2}{b}\right] + i^2\mathbb{E}\left[\|\nabla f(\theta_{t,0})\|^2\right]\right].
$$

Steps (a) and (b) follow from:

$$
\sum_{i=0}^{k-1}\sum_{j=1}^{b}\mathbb{E}\left[\|\nabla f(\theta_{t,i};\xi_{i,j}) - \nabla f(\theta_{t,0})\|^2\right] \le \frac{B}{B}\sum_{i=0}^{\lfloor\frac{B}{b}\rfloor-1}\sum_{j=1}^{b}\mathbb{E}\left[\|\nabla f(\theta_{t,i};\xi_{i,j}) - \nabla f(\theta_{t,0})\|^2\right] \le B\left[\ominus\mathbb{E}\left[\nabla f(\theta_{t,0})\right] + \frac{\sigma^2}{b}\right].
$$

$$
(41)
$$

Consequently:

$$
\sum_{i=0}^{\lfloor \frac{\mathbb{B}}{b} \rfloor - 1} \mathbb{E}\left[\|\theta_{t,i} - \theta_{t,0}\|^2\right] \leq \frac{4\eta_t^2 \lambda^2}{(1-\lambda)^2} \sum_{i=0}^{\lfloor \frac{\mathbb{B}}{b} \rfloor - 1} \left[(1-\lambda)^2 \frac{2Bi}{b}\left[\ominus\mathbb{E}\left[\nabla f(\theta_{t,0})\right] + \frac{\sigma^2}{b}\right] + 2\mathbb{E}\left[\|\nabla f(\theta_{t,0})\|^2\right]\right]
$$

$$
+ 6\eta_t^2 L^2 \sum_{i=0}^{\lfloor \frac{\mathbb{B}}{b} \rfloor - 1} i \sum_{k=0}^{\lfloor \frac{\mathbb{B}}{b} \rfloor - 1} \mathbb{E}\left[\|\theta_{t,k} - \theta_{t,0}\|^2\right]
$$

$$
+ 6\eta_t^2 \left[\sum_{i=0}^{\lfloor \frac{\mathbb{B}}{b} \rfloor - 1} i\frac{B}{b}\left[\ominus\mathbb{E}\left[\nabla f(\theta_{t,0})\right] + \frac{\sigma^2}{b}\right] + \sum_{i=0}^{\lfloor \frac{\mathbb{B}}{b} \rfloor - 1} i^2 \mathbb{E}\left[\|\nabla f(\theta_{t,0})\|^2\right]\right]
$$

$$
\stackrel{\text{(c)}}{=} \frac{4\eta_t^2 \lambda^2}{(1-\lambda)^2}\left[(1-\lambda)^2 \frac{B^3}{b^3}\left[\ominus\mathbb{E}\left[\nabla f(\theta_{t,0})\right] + \frac{\sigma^2}{b}\right] + 2\frac{B}{b}\mathbb{E}\left[\|\nabla f(\theta_{t,0})\|^2\right]\right]
$$

$$
+ 3\eta_t^2 L^2 \frac{B^2}{b^2} \sum_{k=0}^{\lfloor \frac{\mathbb{B}}{b} \rfloor - 1} \mathbb{E}\left[\|\theta_{t,k} - \theta_{t,0}\|^2\right] + 6\eta_t^2 \left[\frac{B^3}{2b^3}\left[\ominus\mathbb{E}\left[\nabla f(\theta_{t,0})\right] + \frac{\sigma^2}{b}\right] + \frac{B^3}{3b^3}\mathbb{E}\left[\|\nabla f(\theta_{t,0})\|^2\right]\right],
$$

$$(42)$$

where (c) uses $\sum_{i=0}^{\lfloor \frac{\mathbb{B}}{b} \rfloor - 1} i \leq \frac{B^2}{2b^2}$ and $\sum_{i=0}^{\lfloor \frac{\mathbb{B}}{b} \rfloor - 1} i^2 \leq \frac{B^3}{3b^3}$. Defining $S = \sum_{i=0}^{\lfloor \frac{\mathbb{B}}{b} \rfloor - 1} \mathbb{E}\left[\|\theta_{t,i} - \theta_{t,0}\|^2\right]$ and setting $\eta_t \leq \frac{b}{\sqrt{6}BL}$ yields:

$$
S \leq \frac{1}{2}S + \left[\frac{4\eta_t^2 \lambda^2 B^3}{b^3} + \frac{3\eta_t^2 B^3}{b^3}\right]\left[\ominus\mathbb{E}\left[\nabla f(\theta_{t,0})\right] + \frac{\sigma^2}{b}\right] + \left[\frac{8B\eta_t^2 \lambda^2}{b(1-\lambda)^2} + \frac{2\eta_t^2 B^3}{b^3}\right]\mathbb{E}\left[\|\nabla f(\theta_{t,0})\|^2\right],
$$

$$(43)$$

which implies:

$$
S \leq \left[\frac{2\eta_t^2 B^3}{b^3}(4\lambda^2 + 3)\ominus + 4\eta_t^2\left(\frac{4B\lambda^2}{b(1-\lambda)^2} + \frac{B^3}{b^3}\right)\right]\mathbb{E}\left[\|\nabla f(\theta_{t,0})\|^2\right] + \frac{2\eta_t^2 B^3}{b^3}(4\lambda^2 + 3)\frac{\sigma^2}{b}.
$$

$$(44)$$

Under Assumption 1 and noting $z_{t,0} = \theta_{t,0} = \theta_{t-1} = z_{t-1}$:

$$\mathbb{E}\left[f\left(z_t\right)\right] \leq \mathbb{E}\left[f\left(z_{t-1}\right)\right] + \mathbb{E}\left[\left\langle \nabla f\left(z_{t-1}\right), z_t - z_{t-1}\right\rangle\right] + \frac{L}{2}\mathbb{E}\left[\left\|z_t - z_{t-1}\right\|^2\right]$$

$$\leq \mathbb{E}\left[f\left(z_{t-1}\right)\right] + \mathbb{E}\left[\left\langle \nabla f\left(z_{t-1}\right), \frac{\eta_t}{b}\sum_{i=0}^{\lfloor\frac{\mathbb{B}}{b}\rfloor-1}\sum_{j=1}^{b}\nabla f(\theta_{t,i};\xi_{i,j})\right\rangle\right] + \frac{L\eta_t^2}{2b^2}\mathbb{E}\left[\left\|\sum_{i=0}^{\lfloor\frac{\mathbb{B}}{b}\rfloor-1}\sum_{j=1}^{b}\nabla f(\theta_{t,i};\xi_{i,j})\right\|^2\right]$$

$$= \mathbb{E}\left[f\left(z_{t-1}\right)\right] + \frac{L\eta_t^2}{2b^2}\mathbb{E}\left[\left\|\sum_{i=0}^{\lfloor\frac{\mathbb{B}}{b}\rfloor-1}\sum_{j=1}^{b}\nabla f(\theta_{t,i};\xi_{i,j})\right\|^2\right] - \frac{\eta_t B}{2b}\mathbb{E}\left[\left\|\nabla f\left(z_{t-1}\right)\right\|^2\right]$$

$$- \frac{\eta_t B}{2b}\mathbb{E}\left[\left\|\frac{1}{B}\sum_{i=0}^{\lfloor\frac{\mathbb{B}}{b}\rfloor-1}\sum_{j=1}^{b}\nabla f(\theta_{t,i};\xi_{i,j})\right\|^2\right] + \frac{\eta_t B}{2b}\mathbb{E}\left[\left\|\frac{1}{B}\sum_{i=0}^{\lfloor\frac{\mathbb{B}}{b}\rfloor-1}\sum_{j=1}^{b}[\nabla f(\theta_{t,0};\xi_{i,j}) - \nabla f(\theta_{t,i};\xi_{i,j})]\right\|^2\right]$$

$$\leq \mathbb{E}\left[f\left(z_{t-1}\right)\right] + \frac{L\eta_t^2}{2b^2}\mathbb{E}\left[\left\|\sum_{i=0}^{\lfloor\frac{\mathbb{B}}{b}\rfloor-1}\sum_{j=1}^{b}\nabla f(\theta_{t,i};\xi_{i,j})\right\|^2\right] - \frac{\eta_t B}{2b}\mathbb{E}\left[\left\|\nabla f\left(z_{t-1}\right)\right\|^2\right]$$

$$- \frac{\eta_t}{2Bb}\mathbb{E}\left[\left\|\sum_{i=0}^{\lfloor\frac{\mathbb{B}}{b}\rfloor-1}\sum_{j=1}^{b}\nabla f(\theta_{t,i};\xi_{i,j})\right\|^2\right] + \frac{\eta_t B}{2b}\frac{L^2}{B}\sum_{i=0}^{\lfloor\frac{\mathbb{B}}{b}\rfloor-1}\sum_{j=1}^{b}\mathbb{E}\left[\left\|[\theta_{t,0} - \theta_{t,i}]\right\|^2\right]$$

$$= \mathbb{E}\left[f\left(z_{t-1}\right)\right] + \left(\frac{L\eta_t^2}{2b^2} - \frac{\eta_t}{2Bb}\right)\mathbb{E}\left[\left\|\sum_{i=0}^{\lfloor\frac{\mathbb{B}}{b}\rfloor-1}\sum_{j=1}^{b}\nabla f(\theta_{t,i};\xi_{i,j})\right\|^2\right] - \frac{\eta_t B}{2b}\mathbb{E}\left[\left\|\nabla f\left(z_{t-1}\right)\right\|^2\right]$$

$$+ \frac{\eta_t L^2}{2}\sum_{i=0}^{\lfloor\frac{\mathbb{B}}{b}\rfloor-1}\mathbb{E}\left[\left\|[\theta_{t,0} - \theta_{t,i}]\right\|^2\right].$$

Setting $\eta_t = \frac{\eta b}{B}$ with $\eta_t \leq \frac{1}{L}$ implies $\left(\frac{L\eta_t^2}{2b^2} - \frac{\eta_t}{2Bb}\right) \leq 0$. Thus:

$$\mathbb{E}\left[f\left(z_t\right)\right] \leq \mathbb{E}\left[f\left(z_{t-1}\right)\right] - \frac{\eta_t B}{2b}\mathbb{E}\left[\left\|\nabla f\left(z_{t-1}\right)\right\|^2\right] + \frac{\eta_t L^2}{2}\sum_{i=0}^{\lfloor\frac{\mathbb{B}}{b}\rfloor-1}\mathbb{E}\left[\left\|[\theta_{t,0} - \theta_{t,i}]\right\|^2\right]. \quad (45)$$

By combining (44) and (45):, we establish:

$$\mathbb{E}\left[f\left(z_t\right)\right] \leq \mathbb{E}\left[f\left(z_{t-1}\right)\right] - \frac{\eta_t B}{2b}\mathbb{E}\left[\left\|\nabla f\left(z_{t-1}\right)\right\|^2\right] + \frac{\eta_t}{2}\sum_{i=0}^{\lfloor\frac{\mathbb{B}}{b}\rfloor-1}\mathbb{E}\left[\left\|[\theta_{t,0} - \theta_{t,i}]\right\|^2\right]$$

$$\leq \mathbb{E}\left[f\left(z_{t-1}\right)\right] - \frac{\eta_t B}{2b}\mathbb{E}\left[\left\|\nabla f\left(z_{t-1}\right)\right\|^2\right] + \left[\frac{\eta_t^3 L^2 B^3}{b^3}(4\lambda^2 + 3) \ominus + 2\eta_t^3 L^2\left(\frac{4B\lambda^2}{b(1-\lambda)^2} + \frac{B^3}{b^3}\right)\right]$$

$$\cdot \mathbb{E}\left[\left\|\nabla f(\theta_{t,0})\right\|^2\right] + \frac{\eta_t^3 L^2 B^3}{b^3}(4\lambda^2 + 3)\frac{\sigma^2}{b}$$

$$\leq \mathbb{E}\left[f\left(z_{t-1}\right)\right] - \frac{\eta}{2}\mathbb{E}\left[\left\|\nabla f\left(z_{t-1}\right)\right\|^2\right] + \left[\eta^3 L^2 \ominus + 2\eta^3 L^2\left(\frac{4b^2\lambda^2}{B^2(1-\lambda)^2} + 1\right)\right]\mathbb{E}\left[\left\|\nabla f(\theta_{t,0})\right\|^2\right]$$

$$+ \eta^3 L^2(4\lambda^2 + 3)\frac{\sigma^2}{b}$$

$$= \mathbb{E}\left[f\left(\theta_{t-1}\right)\right] - \frac{\eta}{2}\mathbb{E}\left[\left\|\nabla f\left(\theta_{t-1}\right)\right\|^2\right] + \left[\eta^3 L^2 \ominus + 2\eta^3 L^2\left(\frac{4b^2\lambda^2}{B^2(1-\lambda)^2} + 1\right)\right]\mathbb{E}\left[\left\|\nabla f(\theta_{t-1})\right\|^2\right]$$

$$+ \eta^3 L^2(4\lambda^2 + 3)\frac{\sigma^2}{b},$$

next we set $\eta^2 \leq \frac{1}{4L^2(\ominus + 2C)}$, where $C = \frac{4b^2\lambda^2}{B^2(1-\lambda)^2} + 1$, simplifies to :

$$\mathbb{E}\left[f\left(z_t\right)\right] \leq \mathbb{E}\left[f\left(z_{t-1}\right)\right] - \frac{\eta}{4}\mathbb{E}\left[\|\nabla f\left(z_{t-1}\right)\|^2\right] + \eta^3 L^2(4\lambda^2 + 3)\frac{\sigma^2}{b}, \tag{46}$$

by rearranging terms yields:

$$\mathbb{E}\left[\|\nabla f\left(\theta_{t-1}\right)\|^2\right] \leq \frac{4}{\eta}\left[\mathbb{E}\left[f\left(\theta_{t-1}\right)\right] - \mathbb{E}\left[f\left(\theta_t\right)\right]\right] + 4\eta^2 L^2(4\lambda^2 + 3)\frac{\sigma^2}{b}, \tag{47}$$

then summing from $t = 1$ to $T$:

$$\sum_{t=1}^{T}\mathbb{E}\left[\|\nabla f\left(\theta_{t-1}\right)\|^2\right] \leq \frac{4}{\eta}\left[\mathbb{E}\left[f\left(\theta_0\right)\right] - \mathbb{E}\left[f\left(\theta_t\right)\right]\right] + 4\eta^2 L^2 T(4\lambda^2 + 3)\frac{\sigma^2}{b}$$

$$\leq \frac{4}{\eta}\left[\mathbb{E}\left[f\left(\theta_0\right)\right] - \mathbb{E}\left[f^*\right]\right] + 4\eta^2 L^2 T(4\lambda^2 + 3)\frac{\sigma^2}{b} \tag{48}$$

$$= \frac{4}{\eta}\left[f\left(\theta_0\right) - f^*\right] + 4\eta^2 L^2 T(4\lambda^2 + 3)\frac{\sigma^2}{b}.$$

Finally, we obtain:

$$\frac{1}{T}\sum_{t=1}^{T}\mathbb{E}\left[\|\nabla f\left(\theta_{t-1}\right)\|^2\right] \leq \frac{4}{\eta T}\left[f\left(\theta_0\right) - f^*\right] + 4\eta^2 L^2(4\lambda^2 + 3)\frac{\sigma^2}{b}, \tag{49}$$

This completes the proof. $\qquad\square$

