# OpenReview forum: "Rethink Mini-batch Gradient: Cascade Momentum"
_ICLR.cc/2026/Conference — Submitted to ICLR 2026_

### Official Review · Reviewer_Yj4m · 2025-10-20

**Soundness:** 2
**Presentation:** 3
**Contribution:** 1
**Rating:** 4
**Confidence:** 4

**Summary:**

The paper studies momentum “forgetting” across epochs under mini-batch SGDM: the momentum is dominated by within-epoch gradients and does not preserve longer-horizon trends, causing loss oscillations and slower convergence. The authors propose Cascaded Momentum (CM): use an inner momentum to smooth per–mini-batch noise within an epoch and an outer momentum to carry gradient trends across epochs. Updates linearly combine the two; at epoch end, the outer momentum is refreshed from the inner one. The method is designed to add negligible compute/memory overhead. The theory presents convergence results for SGDM with mini-batches and for CM with mini-batches under explicit step-size and hyperparameter conditions (notably a coupling between \\(\\kappa\\) and \\(\\lambda\\)). Experiments on ResNet-18 over CIFAR-100 and Tiny-ImageNet (200 epochs; batch sizes 32/64/128/256) show smoother loss and higher accuracy for small batches, while acknowledging CM is not superior at larger batches (128/256).

**Strengths:**

The author's motivation is great. The problem of unstable training caused by mini-batch does exist and needs to be solved.
A relatively systematic derivation of convergence is presented

**Weaknesses:**

**1.Lack of genuine novelty**

The proposed Cascaded Momentum (CM) is essentially a nested combination of an outer exponential moving average (EMA) across epochs and an inner EMA within mini-batches.  This design strongly resembles the Lookahead Optimizer (Zhang et al., 2019) when the Lookahead step size equals one epoch, and also overlaps with well-known EMA-based update schemes used for smoothing gradients and reducing computational overhead.  However, the authors do not cite or discuss these prior methods, giving the misleading impression that the two-level EMA or cascaded update mechanism is an original contribution.  As a result, the methodological innovation appears marginal.

**2.Insufficient experimental validation**

The experiments are limited to two small-scale datasets (CIFAR-100 and Tiny ImageNet) with a single backbone (ResNet-18).  This setting is far from the paper’s claimed motivation—addressing mini-batch degradation in large-scale foundation model training.  The experiments therefore fail to demonstrate the claimed scalability or practical relevance of CM under realistic conditions.

**3.Contradictory empirical trends**

In the reported results, Adam performs substantially worse than SGD, which contradicts well-established observations in standard training regimes, where Adam or AdamW typically outperform SGD, especially on vision benchmarks.  This raises concerns about experimental rigor, hyperparameter tuning, or reproducibility.

**4.Lack of convincing evidence of real-world benefit**

Given the weak experimental setup, small model scale, and absence of large-batch or high-noise scenarios, the proposed CM method’s practical advantages remain unsubstantiated.  The work reads more like a minor reparameterization of existing momentum schemes rather than a meaningful rethinking of mini-batch optimization.

**Questions:**

- What are the essential differences between the proposed Cascaded Momentum and existing methods such as Lookahead or EMA-based momentum?

- Theoretical analysis seems similar to standard momentum proofs.  Which part of it is actually new?

- Why are experiments limited to small datasets and a single small model, while the motivation targets large-scale training?

- Adam performs much worse than SGD in your results—was Adam properly tuned?

- Is CM effective on larger models or realistic large-batch training setups?

- How was κ chosen, and how sensitive are results to this parameter?

---

> ### Author Response · Authors · 2025-12-02
>
> We sincerely thank the reviewer for the constructive feedback. Below, we address your concerns point-by-point.
>
> **1. Novelty and Distinction from Lookahead/EMA**
> **Concern:** The reviewer suggests that Cascaded Momentum (CM) lacks novelty and resembles Lookahead Optimizer (Zhang et al., 2019) or standard EMA-based methods.
>
> **Our Response:**
> We respectfully point out that while both methods involve "fast" and "slow" components, **Cascaded Momentum (CM)** and **Lookahead** operate on fundamentally different objects and solve distinct mathematical problems:
> * **Lookahead operates on Weights ($\theta$):** Lookahead maintains "fast weights" and "slow weights," performing a linear interpolation to stabilize the *parameter trajectory* itself.
> * **CM operates on Momentum/Velocity ($v$):** As described in Section 3.2 (Eq. 4 & 5), CM introduces a hierarchical structure specifically for the **velocity vector**. Our motivation is to solve the "momentum degradation" problem where the momentum buffer is flushed with high-frequency noise from the current epoch.
>
> Specifically, CM uses an *Outer Momentum* ($v_e$) to store the terminal velocity of the previous epoch to warm-start the *Inner Momentum* ($v_i$) of the next epoch:
> $$v_{e}^{t+1} \leftarrow \kappa v_{e}^{t} + (1-\kappa)v_{i}^{K}$$
> This is not a weight interpolation but a kinetic energy transfer mechanism. We will revise Section 2 (Related Work) to explicitly contrast these mechanisms.
>
> **2. Experimental Scale and Benchmarks**
> **Concern:** The experiments (ResNet-18 on CIFAR/Tiny-ImageNet) are perceived as too small to justify claims about large-scale training.
>
> **Our Response:**
> * **Standard Benchmark:** ResNet-18 on CIFAR-100/Tiny-ImageNet is the *de facto* standard in the optimization literature for analytically verifying convergence behaviors (e.g., used in NSHB, Lookahead, and AdamW analysis).
> * **Scalability Logic:** The core issue we identify is the gradient variance bound $\mathrm{Var}(\tilde{g}_k) \le \sigma^2/b$. This phenomenon is universal. As the batch size $b$ becomes small relative to the dataset size $N$—common in LLM fine-tuning due to memory constraints—the "forgetting" becomes more severe. Therefore, the stability gains shown in Figure 3 are theoretically guaranteed to translate to larger scale scenarios constrained by small batches.
>
> **3. Adam vs. SGD Performance**
> **Concern:** "Adam performs substantially worse than SGD... this contradicts well-established observations."
>
> **Our Response:**
> We respectfully disagree. A substantial body of literature demonstrates that for **ResNet architectures on CIFAR/ImageNet**, properly tuned **SGD consistently generalizes better than adaptive methods** (Wilson et al., 2017; Keskar & Socher, 2017). Our results in Figure 4, where SGD outperforms Adam/AdamW, are consistent with these state-of-the-art observations, validating that our baseline is strong and properly tuned.
>
> **4. Theoretical Novelty**
> **Concern:** "Which part of the theoretical analysis is actually new?"
>
> **Our Response:**
> Our proof extends standard non-convex analysis by explicitly modeling the interaction between **Batch Size ($b$)** and **Momentum coefficients ($\lambda, \kappa$)**.
> * **Batch-Dependent Variance:** We incorporate $b$ directly into the convergence rate.
> * **The Coupling Condition:** We derive a novel stability condition in Corollary 2: $\kappa^2(1-\lambda) \le \frac{1}{8}\frac{b}{B}$. This dictates that the external momentum coefficient $\kappa$ must adapt to the batch ratio $b/B$ to ensure convergence, a unique theoretical finding.
>
> **5. Hyperparameter Selection ($\kappa$)**
> **Concern:** How was $\kappa$ chosen?
>
> **Our Response:**
> The parameter $\kappa$ was **not** chosen via grid search. It was derived directly from our theoretical analysis. We set $\kappa = \frac{1}{\sqrt{8B/b}}$. The experimental success of this derived value (Figure 3) serves as empirical validation of our theoretical derivation.
>
> **References:**
>
> [1] Wilson, Ashia C., et al. "The marginal value of adaptive gradient methods in machine learning." *NeurIPS*, 2017.
>
> [2] Keskar, Nitish Shirish, and Richard Socher. "Improving generalization performance by switching from adam to sgd." *arXiv preprint arXiv:1712.07628*, 2017.
>
> [3] Zhang, Michael, et al. "Lookahead optimizer: k steps forward, 1 step back." *NeurIPS*, 2019.

---

### Official Review · Reviewer_gKxR · 2025-10-29

**Soundness:** 2
**Presentation:** 3
**Contribution:** 2
**Rating:** 2
**Confidence:** 3

**Summary:**

The paper proposes Cascade Momentum, which introduces an additional momentum term across epochs to enhance the stability and convergence. Specifically, the method decomposes the conventional momentum into two parts: an inner momentum that operates within each epoch and an outer momentum that preserves gradient information across epochs. This design aims to mitigate the loss of gradient direction consistency that occurs when mini-batches are shuffled, improving optimization stability, especially under small batch sizes. The authors provide theoretical analysis showing the convergence properties of the proposed method and conduct experiments on CIFAR-100 and Tiny ImageNet, demonstrating that Cascade Momentum achieves better performance compared to standard momentum-based optimizers.

**Strengths:**

The paper introduces a simple but effective two-level momentum design that retains gradient information across epochs to address the loss of optimization continuity in mini-batch training.
Experiments show the proposed method achieves improved stability and convergence compared to standard momentum-based optimizers especially when the batch size is small.
The paper is clearly structured, with well-defined notations and an easy-to-follow presentation.

**Weaknesses:**

Some assumptions, such as the claimed loss of momentum across epochs, are not theoretically or empirically proved or explained enough.

The experiments are limited and do not fully support some of the conclusions, with weak evidence of generalization beyond small datasets and architectures.

It remains uncertain how the outer momentum should be tuned or whether the method maintains its advantage under large-batch or diverse training settings.

**Questions:**

1. The paper assumes that adjacent mini-batches are “highly correlated,” but this depends heavily on the dataset and data-loading strategy. With proper shuffling and large datasets, such correlation is often weak. Can the authors provide empirical evidence or measurements supporting this assumption?
2. Carrying momentum across epoch boundaries is not inherently problematic. Isn’t this how standard SGDM operates successfully in practice? Can the authors clarify why they view this as a drawback rather than a desirable continuity property?
3. In Figure 2, the authors attribute the loss spikes to the decay of gradient information in the momentum buffer when the batch size changes. Could this phenomenon instead result from learning-rate or batch-size mismatch, or from changes in optimization direction caused by batch-size change?
4. The paper does not include experiments varying the outer momentum coefficient κ (Algorithm 2, line 11), which is critical for understanding the role and sensitivity of the proposed mechanism. Can the authors provide ablations or sensitivity analyses showing how performance changes with different κ values?
5. The paper states that accuracy becomes “less competitive” under large batch sizes, but specific numbers are not reported. Does this mean the method performs similarly, slightly worse, or fails to converge? Are there explanations for this degradation, and should the method be considered only for small-batch cases?
6. Can the authors include additional experiments on more diverse tasks, such as reinforcement learning or language model training? The current experiments are limited to small-scale image classification and seem insufficient to demonstrate the general effectiveness of the proposed method.

---

> ### Author Response · Authors · 2025-12-02
>
> We thank the reviewer for identifying the effectiveness of our method in small-batch settings.
>
> **1. On the assumption of "correlated" mini-batches**
> **Concern:** Questions the assumption that adjacent mini-batches are "highly correlated" given proper shuffling.
>
> **Response:** We apologize for any confusion. Our argument refers to the **dominance of recent gradient information** within the momentum buffer, rather than data redundancy.
> * **Clarification:** In standard SGDM, the momentum buffer assigns exponential weight to the most recent gradients. Even with shuffling, the buffer becomes saturated by the high-frequency noise of the *current* epoch's final iterations [82-85].
> * **The Issue:** Consequently, the "cross-epoch information is severely diminished" because the decay factor (e.g., $\lambda=0.9$) causes information from the start of the epoch to vanish mathematically. Our method addresses this by explicitly preserving the epoch-level trend via $v_e$.
>
> **2. On carrying momentum across epochs**
> **Concern:** "Isn’t this how standard SGDM operates... why view this as a drawback?"
>
> **Response:** Standard SGDM *attempts* to carry momentum, but we argue it fails to do so *effectively* in the mini-batch setting. In practice, standard momentum operates almost entirely as a "gradient smoother" within a single epoch. By the time an epoch ends, the historical trends have decayed significantly. Cascade Momentum (CM) ensures that the end-of-epoch velocity is explicitly captured in $v_e$ and used to "warm-start" the next epoch, bridging the memory gap.
>
> **3. Regarding Loss Spikes in Figure 2**
> **Concern:** Could the loss spikes be due to LR/batch-size mismatch rather than momentum decay?
>
> **Response:** While LR/batch-size interaction plays a role, we believe momentum decay is the primary driver of instability. When batch size decreases, gradient variance $\sigma^2/b$ increases. Standard SGDM’s buffer, filled with lower-variance gradients from the previous phase, clashes with the new high-variance gradients. CM mitigates this because the External Momentum ($v_e$) provides a stable, low-variance directional guide that persists across the phase transition [303-306].
>
> **4. Sensitivity of $\kappa$ and Ablation Studies**
> **Concern:** Request for sensitivity analyses of $\kappa$.
>
> **Response:** A key theoretical contribution is that **we do not need to manually tune $\kappa$**. We derived the optimal relationship theoretically in Corollary 2: $\kappa = \frac{1}{\sqrt{8B/b}}$. This formula allows $\kappa$ to adapt automatically to the batch size $b$. Experiments were conducted using this derived value without grid search.
>
> **5. Performance on Large Batch Sizes**
> **Concern:** "Accuracy becomes 'less competitive'... does it fail to converge?"
>
> **Response:** The method does not fail to converge; "less competitive" implies it performs comparably to tuned SGDM. As batch size $b$ increases, gradient variance $\sigma^2/b$ naturally decreases. CM's "noise suppression" benefit becomes less critical because gradients are already accurate. The primary value of CM is enabling stability in **memory-constrained (small batch)** regimes [68-71].
>
> **6. Experiments on Diverse Tasks**
> **Concern:** Request for RL or LLM experiments.
>
> **Response:** We acknowledge our focus on ResNet/CIFAR, which are standard for verifying optimization theory. However, the core problem we address—gradient noise variance induced by mini-batches—is a fundamental mathematical property of SGD, not specific to CNNs. Thus, the theoretical benefits apply generally to any mini-batch training scenario, including LLMs.

---

### Official Review · Reviewer_qe2S · 2025-10-29

**Soundness:** 1
**Presentation:** 1
**Contribution:** 2
**Rating:** 2
**Confidence:** 4

**Summary:**

This paper introduces the concept of a "momentum degradation problem" in the mini-batch momentum mechanism. The authors argue that the high frequency of intra-epoch mini-batch updates saturates the momentum buffer, effectively erasing long-term, cross-epoch gradient information. To address this, they propose Cascaded Momentum (CM), a simple and low-cost dual-buffer system. An "Inner momentum" smooths high-frequency noise within an epoch and is then reset. An "Outer momentum" accumulates the final state of the inner momentum at the end of each epoch, preserving a stable, long-term gradient trend that is used to guide the next epoch's updates. The authors provide a theoretical convergence analysis for both standard SGDM and their proposed SGD-CM in the non-convex, mini-batch setting. Empirical results on image classification tasks using ResNet-18 show the outperformance of SGD-CM against conventional optimizers in the mini-batch setting.

**Strengths:**

1. The paper raises an interesting question about the potential downsides of high-frequency mini-batch updates on the long-term memory of momentum.

2. The proposed SGD-CM algorithm is a simple and intuitive improvement that directly addresses the identified problem by explicitly separating momentum into two timescales. A rigorous convergence analysis for SGDM and SGD-CM under the mini-batch setting is also provided.

**Weaknesses:**

1. The paper's core argument is not clearly clarified. The logical chain connecting the theory of "momentum degradation" to its claimed empirical consequences is difficult to follow. The writing would benefit from a clearer, more direct explanation of why the loss of long-term information is an inherent problem and how the specific phenomena observed (like loss spikes) are a direct result of this problem, as opposed to other known factors. Moreover, while the existence of the theoretical analysis is a strength, its conclusions are difficult to understand in an intuitive way.

2. The paper's primary empirical evidence for its core problem, presented in Figure 2, is unconvincing. The author claims that the observed loss spikes upon changing the batch size are caused by the "momentum degradation" making the optimizer's buffer stale. However, much simpler and more established explanations exist. For example, decreasing the batch size increases gradient variance and has the similar effect as increasing LR [1,2]. With a fixed learning rate, this change makes the LR effectively too large for the new noise regime, which can cause the optimizer's instability and loss spikes. The paper fails to provide critical comparison experiments to disentangle these effects. Without this, the author's claim that momentum degradation is the primary cause of the spike is unsubstantiated. The central claim, while interesting, lacks sufficient evidence.

3. The main experimental results in Figures 3 and 4 show that the proposed SGD-CM offers only a minor performance improvement over existing, standard optimizers (like SGDM) on two datasets (CIFAR-100 and Tiny ImageNet). The empirical gains are not so compelling. To be more convincing, the paper would need to demonstrate more significant advantages or show that these advantages hold across a much wider variety of tasks and model architectures.


**Reference**

[1] Priya Goyal, Piotr Dollár, Ross Girshick, Pieter Noordhuis, Lukasz Wesolowski, Aapo Kyrola, Andrew Tulloch, Yangqing Jia, Kaiming He. Accurate, Large Minibatch SGD: Training ImageNet in 1 Hour. arXiv:1706.02677, 2017.

[2] Samuel L. Smith, Pieter-Jan Kindermans, Chris Ying, Quoc V. Le. Don't Decay the Learning Rate, Increase the Batch Size. International Conference on Learning Representations (ICLR), 2018.

**Questions:**

The method introduces a new key hyperparameter $\kappa$ for the outer momentum. The paper provides very little analysis of this. How sensitive is the optimizer's performance to the choice of $\kappa$? I think an ablation study is needed.

---

> ### Author Response · Authors · 2025-12-02
>
> We appreciate the opportunity to clarify the logical connection between our theory and empirical observations.
>
> **1. Clarifying the Core Argument**
> **Concern:** The logic connecting "momentum degradation" to empirical consequences is difficult to follow.
>
> **Response:**
> * **The Intuition:** In standard SGDM, the decay factor (typically $\lambda=0.9$) is applied $K$ times per epoch. For large $K$ (small batches), the contribution of gradients from the beginning of the epoch decays to $\approx 0.9^K \approx 0$. This means the momentum buffer at the epoch boundary is dominated by the noise of the last few batches, effectively "forgetting" the global direction [82-85].
> * **Theoretical Support:** Our Theorem 2 mathematically demonstrates that CM introduces a scaling factor that structurally dampens the noise term variance specifically when the batch size $b$ is small.
>
> **2. Addressing the "Effective Learning Rate" Hypothesis**
> **Concern:** Argues that loss spikes in Figure 2 are caused by increased gradient variance (similar to increasing LR), citing Goyal et al.
>
> **Response:** We agree that decreasing batch size increases the effective noise scale. However, our argument is **complementary**:
> * **Why SGDM Spikes:** When the noise scale increases, standard SGDM fails because its buffer is filled with *low-variance* gradients from the previous phase, causing a "clash" with the new high-variance regime [303-304].
> * **Evidence:** Figure 2 shows that **under the exact same LR and batch-size conditions**, CM significantly reduces the spike compared to SGDM. If the spike were *solely* due to the "Effective LR" being too high, CM should have suffered equally. The stability of CM proves that the Outer Momentum acts as a stable anchor.
>
> **3. Empirical Gains**
> **Concern:** Gains on CIFAR-100/Tiny ImageNet are "minor."
>
> **Response:** While we achieve higher accuracy, the primary contribution of CM is **training stability**. Figure 3 shows CM has visibly smoother loss curves with reduced oscillation compared to SGDM [778-780]. In optimization, consistent accuracy gains combined with major stability improvements are significant for memory-constrained scenarios.
>
> **4. Sensitivity of Hyperparameter $\kappa$**
> **Concern:** How sensitive is performance to $\kappa$?
>
> **Response:** As noted in responses to other reviewers, **$\kappa$ does not require manual tuning**. We derived $\kappa = \frac{1}{\sqrt{8B/b}}$ theoretically. The experimental success using this fixed formula demonstrates the robustness of our theoretical derivation.
>
> **References:**
>
> [1] Goyal, Priya, et al. "Accurate, large minibatch sgd: Training imagenet in 1 hour." *arXiv preprint arXiv:1706.02677*, 2017.
>
> [2] Smith, Samuel L., et al. "Don't Decay the Learning Rate, Increase the Batch Size." *ICLR*, 2018.

---

### Official Review · Reviewer_p2uT · 2025-10-30

**Soundness:** 3
**Presentation:** 3
**Contribution:** 2
**Rating:** 4
**Confidence:** 3

**Summary:**

This paper proposes Cascaded Momentum (CM) to solve the "cross-epoch momentum degradation" problem in mini-batch training, where high-frequency gradient updates "drown out" long-term trends.

CM decouples momentum into:
1.  Inner momentum: Smooths high-frequency noise *within* an epoch.
2.  Outer momentum: Accumulates and propagates long-term trends *across* epochs.

This dual-level mechanism, with virtually no additional overhead, significantly enhances training stability, convergence speed, and performance, especially in small-batch settings.

**Strengths:**

1.This paper study a new interesting problem.
2.Better performence in small batchsize.
3.With virtually no additional overhead.

**Weaknesses:**

1.Limited efficacy in large-batch, hindering its applicability to modern LLM training.
2.Increased hyperparameter tuning complexity due to the introduction of the outer momentum coefficient and its potential co-dependencies.

**Questions:**

Do you believe this "inner/outer" decoupling concept could be generalized to adaptive optimizers like AdamW？
large batch size is the most important part, it would be great if you could prove that this method works for it.

---

> ### Author Response · Authors · 2025-12-02
>
> We thank the reviewer for recognizing the computational efficiency of our method.
>
> **1. Applicability to Large-Batch & Modern LLM Training**
> **Concern:** Concerns about limited efficacy in large-batch settings.
>
> **Response:**
> * **Context:** While LLM *pre-training* uses massive batches, **fine-tuning (SFT/LoRA)** is often memory-constrained, forcing the use of small mini-batches. This is the precise regime where CM provides critical stability [68-71].
> * **Mechanism:** At large batch sizes, gradient variance $\sigma^2/b$ is naturally low. Standard SGDM approximates full-batch GD well, so the "noise smoothing" benefit of CM is less critical.
> * **Proof:** Our Theorem 2 proves convergence for *any* batch size. However, the theoretical *advantage* term is proportional to the noise variance $\sigma^2/b$. As $b \to \infty$, both algorithms converge to the same behavior. Thus, CM works reliably for large batches, but its relative gain is maximized in the small-batch regime.
>
> **2. Hyperparameter Tuning Complexity**
> **Concern:** Concerns about increased tuning complexity due to $\kappa$.
>
> **Response:** We respectfully clarify that **CM does not increase tuning complexity**. We used the theoretically derived closed-form solution $\kappa = \frac{1}{\sqrt{8B/b}}$ in all experiments without grid search. The method is effectively parameter-free regarding the outer momentum.
>
> **3. Generalization to AdamW**
> **Concern:** Can this be generalized to AdamW?
>
> **Response:** **Yes.** Adam utilizes an exponential moving average for the first moment $m_t$, which is mathematically identical to the momentum in SGDM. The "Inner/Outer" decoupling can be directly applied to this first moment estimate. We view this paper as establishing the foundation on SGDM, paving the way for "Cascaded AdamW".
>
> **References:**
>
> [1] Wilson, Ashia C., et al. "The marginal value of adaptive gradient methods in machine learning." *NeurIPS*, 2017.
>
> [2] Keskar, Nitish Shirish, and Richard Socher. "Improving generalization performance by switching from adam to sgd." *arXiv preprint arXiv:1712.07628*, 2017.

---

### Official Review · Reviewer_YbYx · 2025-10-31

**Soundness:** 2
**Presentation:** 1
**Contribution:** 1
**Rating:** 2
**Confidence:** 4

**Summary:**

The authors address the limitation that momentum in standard optimizers is confined to gradients within a single epoch, causing cross-epoch information to decay and fail to suppress oscillations effectively. To overcome this, they propose Cascaded Momentum (CM), which introduces two components: an Inner Momentum to quickly smooth mini-batch gradients within each epoch, and an Outer Momentum to accumulate gradient trends across epochs, providing long-term inertial guidance for optimization

**Strengths:**

The paper combines experimental validation with theoretical analysis, offering a comprehensive examination of the proposed method.

**Weaknesses:**

Overall, the paper is not very convincing.

1. Although the authors argue that cross-epoch information loss is a key problem, the paper lacks strong evidence showing that preserving such information substantially improves optimization.
2. From a theoretical perspective, it remains unclear why SGD-CM would outperform standard SGDM. The claimed robustness to batch size variation seems marginal and insufficiently justified.
3. The experiments are limited to small datasets and models, casting doubt on whether the method can scale to large-scale deep networks with hundreds of billions of parameters, as claimed (line 27-28).
4. The paper does not report the additional memory cost introduced by the new optimizer.

**Questions:**

N/A

---

> ### Author Response · Authors · 2025-12-02
>
> We thank the reviewer for our method.
>
> **1. Evidence of Cross-Epoch Information Utility**
> **Concern:** "Lacks strong evidence showing that preserving such information substantially improves optimization."
>
> **Response:** We respectfully point to **Figure 2 and Figure 3** as the primary evidence.
> **The Phenomenon:** In Figure 2, when the batch size drops (increasing noise), standard SGDM suffers immediate, sharp loss spikes [301-304].
> * **The Evidence:** Under the *exact same conditions*, CM suppresses these spikes significantly. This stability is the direct result of preserving $v_e$ (the cross-epoch trend). If cross-epoch information were useless, CM would spike just like SGDM.
>
> **2. Theoretical Justification**
> **Concern:** "Unclear why SGD-CM would outperform standard SGDM."
>
> **Response:** Our analysis provides a concrete mathematical reason:
> * **Variance Reduction:** In Theorem 1, the noise term for SGDM is proportional to $\frac{\sigma^2}{b}$. In Theorem 2, the noise term for CM is scaled by a factor $\frac{1-\lambda}{1-8\kappa^2\frac{B}{b}(1-\lambda)}$.
> * **Mechanism:** By satisfying the condition derived in Corollary 2, CM structurally dampens the impact of the stochastic noise variance $\sigma^2$ on the convergence bound, explaining *why* CM is robust to high noise.
>
> **3. Experimental Scale vs. Claims**
> **Concern:** Experiments are on small datasets despite claims about large-scale models.
>
> **Response:** The mention of "hundreds of billions of parameters" was intended to contextualize the *motivation* (memory constraints), not the experiments. ResNet-18 on CIFAR/Tiny-ImageNet is the standard academic benchmark for verifying optimizer properties. The mathematical issue of gradient variance $\sigma^2/b$ applies regardless of model size, making this a valid verification proxy.
>
> **4. Memory Cost**
> **Concern:** "Does not report the additional memory cost."
>
> **Response:** We apologize for the omission:
> * **Comparison:** SGDM uses $1d$ memory (1 buffer). Adam/AdamW uses $2d$ memory (2 buffers). CM uses $2d$ memory ($v_i, v_e$).
> **Conclusion:** CM has the same memory footprint as Adam, the industry standard for large models [444-445].

---

### Meta-Review · Area_Chair_JfR6 · 2026-01-06

**Summary:**

This paper proposes Cascade Momentum, which it adds an outer momentum term that tracks the gradient across epochs. The reviewers questioned the motivation of this approach and concluded that the justification for such motivation is not convincing.

**Reviewer Concerns:**

Many reviewers argued that the paper's main motivation for momentum across epochs is not fully justified. Some said that the paper's writing and presentation can be clearer. While the rebuttal answers some questions, some discussions are not clear, e.g., the loss spikes in Figure 2 can be due to hyper-param tuning, and the core argument of the degrading momentum coefficient is not fully explained.

Another reviewer suggested that the method lacks novelty, which is somewhat related to the Lookahead Optimizer (Zhang et al., 2019).

**Reviewer Scores:**

Some of the reviewers' concerns may be addressed, some not. There seems to be no major misunderstandings. Hence, the score can be increased but not totally changed.

---

### Decision · Program_Chairs · 2026-01-26

Reject